# Intracellular galectin-3 is a lipopolysaccharide sensor that promotes glycolysis through mTORC1 activation

Xing Chen[1,2,3], Chunyu Yu[2], Xinhua Liu[2], Beibei Liu[1], Xiaodi Wu[1], Jiajing Wu[1], Dong Yan[1], Lulu Han[1], Zifan Tang[1,4], Xinyi Yuan[2], Jianqiu Wang [2], Yue Wang[2,4], Shumeng Liu[1], Lin Shan[1] & Yongfeng Shang [1,2,4] ✉

How the carbohydrate binding protein galectin-3 might act as a diabetogenic and tumorogenic factor remains to be investigated. Here we report that intracellular galectin-3 interacts with Rag GTPases and Ragulator on lysosomes. We show that galectin-3 senses lipopolysaccharide (LPS) to facilitate the interaction of Rag GTPases and Ragulator, leading to the activation of mTORC1. We find that the lipopolysaccharide/galectin-3-Rag GTPases/Ragulator-mTORC1 axis regulates a cohort of genes including *GLUT1*, and *HK2*, and *PKM2* that are critically involved in glucose uptake and glycolysis. Indeed, galectin-3 deficiency severely compromises LPS-promoted glycolysis. Importantly, the expression of HK2 is significantly reduced in diabetes patients. In multiple types of cancer including hepatocellular carcinoma (HCC), galectin-3 is highly expressed, and its level of expression is positively correlated with that of HK2 and PKM2 and negatively correlated with the prognosis of HCC patients. Our study unravels that galectin-3 is a sensor of LPS, an important modulator of the mTORC1 signaling, and a critical regulator of glucose metabolism.

Galectin-3 is a member of the β-galactoside-binding protein family characterized by a carbohydrate recognition domain and proline/glycine-rich tandem repeats[1–3]. Galectin-3 is widely expressed in human tissues and exhibits multifaceted functions dependent on its subcellular compartmentalization. Extracellular galectin-3 is thought to mediate cell adhesion and cell-cell interaction through specific recognition of complex carbohydrates on cell surface[4,5], while intracellular galectin-3 is implicated in cell apoptosis, autophagy, and inflammation[6–9]. Interestingly, recent studies suggest that galectin-3 is involved in cellular metabolism and linked to diabetes and cancer[10–13]. It is reported that galectin-3 deficiency is associated with dysregulated glucose metabolism and exacerbates hyperglycemia[14–17], and it is thus proposed that

galectin-3 is beneficial to glucose homeostasis and has a protective effect on diabetogenesis when nutrients are excess[18,19]. In a number of types of cancer including hepatocellular carcinoma (HCC) and triple-negative breast cancer, galectin-3 is highly expressed and implicated in the development of these cancers[20–22]. Indeed, galectin-3 has been investigating as a potential marker for these malignancies[20–22]. However, the molecular mechanism underlying these pathophysiological activities of galectin-3 is still to be investigated.

The rapamycin complex 1 (mTORC1) Ser/Thr protein kinase is a master regulator of cell growth and organismal homeostasis[23]. In effect, mTORC1 receives various distinct signals from growth factors, amino acids, glucose, cholesterol, and cellular energy status (high ATP/AMP

[1]Department of Biochemistry and Molecular Biology, School of Basic Medical Sciences, Capital Medical University, 100069 Beijing, China. [2]Department of Biochemistry and Molecular Biology, School of Basic Medical Sciences, Hangzhou Normal University, 311121 Hangzhou, China. [3]NHC Key Laboratory of Human Disease Comparative Medicine, National Human Diseases Animal Model Resource Center, Institute of Laboratory Animal Sciences, Chinese Academy of Medical Sciences and Comparative Medicine Center, Peking Union Medical College, 100021 Beijing, China. [4]Department of Biochemistry and Molecular Biology, School of Basic Medical Sciences, Peking University Health Science Center, 100191 Beijing, China. ✉e-mail: yshang@hsc.pku.edu.cn

ratio) to regulate an array of anabolic and catabolic processes through the activation of its downstream targets[23–26]. Accordingly, dysregulation or dysfunction of the mTORC1 signaling pathway are associated with various pathological states, including diabetes and cancer[27].

The molecular events involved in the sensing and integration of intra- and extra-cellular stimuli by mTORC1 are extremely complex. In recent years, great progress has been made in understanding the molecular mechanism by which mTORC1 integrates amino-acid signals on lysosomal surface. It is believed that, in the presence of amino acid, Rag GTPases form obligate heterodimers comprised of RagA or RagB and RagC or RagD to recruit mTOR to lysosomal surface and to activate mTORC1[28–31]. In the process, amino acids regulate the nucleotide state of Rag GTPases dependent on complicated interactions between several distinct factors including Ragulator, composed of p14, p18, MP1, HBXIP, and C7orf59, which serves as a lysosomal scaffold for Rag GTPases and activates RagA or RagB (RagA/B) through its guanine nucleotide exchange factor (GEF)[32–34]. In addition, GATOR1, composed of DEPDC5, NPRL2, and NPRL3, inhibits the mTORC1 signaling[35], while GATOR2, composed of MIOS, SEH1L, WDR24, WDR59, and SEC13, acts upstream of GATOR1 to positively regulate the mTORC1 signaling[35]. Moreover, KICSTOR, composed of KPTN, ITFG2, SZT2, and C12orf66, recruits GATOR1 to lysosomal surface and represents a key component in nutrient stimulation of mTORC1[36,37], while FLCN/FNIP2 plays a critical role in sensing amino-acid levels for the mTORC1 pathway[38,39]. In the past years, several amino-acid sensors have been identified[24,40–42] that greatly advances the understanding of the anatomy of the mTORC1 system. Nevertheless, since the mTORC1 system on lysosomal surfaces is extremely enormous and highly complicated, its metabolic regulation still needs to be delineated.

Lipopolysaccharide (LPS) is a component of the outer wall of the cytoderm of Gram-negative bacteria[43]. The blood level of LPS fluctuates with intestinal microbiota, and high level of LPS is associated with chronic subclinical inflammatory processes, obesity, changes in glucose metabolism and even cancer[44–46]. It is suggested that LPS is inextricably linked to diabetes and cancer, and that elevated LPS level has a protective effect on the development of certain diabetes[47,48]. However, how LPS influences glucose metabolism and is linked to diabetes and cancer remains to be investigated.

We report in the current study that intracellular galectin-3 senses LPS to facilitate the interaction of Rag GTPases and Ragulator on lysosome, resulting in activation of Rag GTPases, lysosomal loading of mTOR, and activation of the mTORC1 signaling. We show that the LPS/galectin-3-Rag GTPases/Ragulator-mTORC1 axis regulates a group of genes including *GLUT1* (glucose transporter member 1), *HK2* (hexokinase 2), and *PKM2* (pyruvate kinase M2) that represent the well-characterized glycolysis-associated genes. We explore the clinicopathological significance of this axis in cellular glycolysis and in diabetes and cancer and reveal a critical role of galectin-3 in diabetogenesis and tumorigenesis.

## Results

### Galectin-3 is physically associated with components of the mTORC1 signaling system on lysosomal surface

As stated above, intracellular galectin-3 has been shown to regulate cell apoptosis, autophagy, and inflammation[6–9]. Interestingly, recent investigations suggest that galectin-3 is involved in cellular metabolism and linked to diabetes and cancer[10–13]. To further explore the biological function of intracellular galectin-3, we first interrogated the in vivo galectin-3 interactome by epitope-based proteomic screening with combined immunopurification and mass spectrometry. To this end, FLAG-tagged galectin-3 (FLAG-Gal3) was stably expressed in human embryonic kidney HEK-293T cells. Cellular extracts were prepared and subjected to affinity purification using an anti-FLAG affinity column. The purified proteins were resolved on SDS-PAGE and analyzed by mass spectrometry. We found that a list of proteins, including

SLC38A9, ATP6V1B2 (v-ATPase component), RagA, RagB, and RagC (Rag GTPases), and p14, p18, and MP1 (Ragulator subunits), all constituents of the mTORC1 machinery, were co-purified with galectin-3 (Fig. 1a, Supplementary Data 1).

To corroborate this observation, the epitope-based proteomic screening was also performed in HEK-293T cells expressing FLAG-RagA or FLAG-RagC. Combined immunopurification and mass spectrometry showed that galectin-3, in addition to RagA, RagB, RagC, p14, p18, and MP1, was indeed also co-purified with both RagA and RagC (Fig. 1a, Supplementary Data 1). Consistently, analysis of the BioPlex protein-protein interaction database generated by immunoprecipitation and mass spectrometry for more than 5,000 proteins stably expressed in HEK-293T cells[49] found that galectin-3 is among the interacting partners of Rag GTPases (RagB and RagC), and the STRING and CF-MS explorer (https://cf-ms-browser.msl.ubc.ca/) protein-protein interaction databases also showed the interaction between galectin-3 and Rag GTPases (RagB and RagC) (Fig. 1b).

To further validate the interaction of galectin-3 with the components of the mTORC1 signaling system, co-immunoprecipitation experiments were carried out first in HEK-293T cells expressing FLAG-Gal3. Immunoprecipitation with anti-FLAG followed by immunoblotting with antibodies against the components of the mTORC1 signaling pathway showed that RagA and RagC, p14 and p18, SLC38A9, and ATP6V1B2 were all efficiently co-immunoprecipitated with galectin-3, whereas no interaction was detected for galectin-3 with DEPDC5 and NPRLl2 (GATOR1 components) or with SEC13, WDR24, and WDR59 (GATOR2 components) (Fig. 1c). Co-immunoprecipitation experiments with antibodies detecting endogenous proteins also confirmed that galectin-3 interacts with RagA/RagC, p14/p18, SLC38A9, and ATP6V1B2 (Fig. 1d). Reciprocally, immunoprecipitation with antibodies against RagA, RagC, p14, p18, ATP6V1B2, or SLC38A9 followed by immunoblotting with antibodies against galectin-3 also showed that galectin-3 was efficiently co-immunoprecipitated with these components of the mTORC1 signaling pathway (Fig. 1d). Moreover, immunofluorescent staining in HEK-293T cells showed that GFP-fused galectin-3 (GFP-Gal3) was co-localized with RagC, p18, and Lamp2 (lysosomal marker) (Fig. 1e), supporting a notion that the interaction of galectin-3 with Rag GTPases and Ragulator occurs on lysosomes.

To further support the physical interaction of galectin-3 with the mTORC1 machinery, FLAG-Gal3 affinity proteins from HEK-293T cells were fractionated by fast protein liquid chromatography (FPLC) with a Superose 6 column and a high salt extraction and size exclusion approach. Galectin-3 was eluted with an apparent molecular mass much greater than that of the monomeric protein; FLAG-Gal3 immunoreactivity was detected in chromatographic fractions with an elution pattern largely overlapped with that of the lysosomal protein Lamp2 and the components of the mTORC1 machinery, including RagA, RagC, p14, p18, SLC38A9, and ATP6V1B2 (Fig. 1f). In addition, glutathione S-transferase (GST) pull-down assays with bacterially-purified GST-galectin-3 (GST-Gal3) and in vitro transcribed/translated components of the mTORC1 system including RagA, RagC, p14, p18, SLC38A9, or ATP6V1B2 showed that galectin-3 was capable of interacting with RagA, RagC, p14, and p18, but not with the other components of the mTORC1 signaling system that we tested (Fig. 1g). GST pull-down experiments with GST-fused RagA, RagC, p14, p18, SLC38A9, or ATP6V1B2 and in vitro transcribed/translated galectin-3 corroborated these observations (Fig. 1g). Moreover, we also performed co-immunoprecipitation experiments in HEK-293T cells expressing FLAG-galectin-2 (FLAG-Gal2) or FLAG-galectin-4 (FLAG-Gal4) to represent the prototype or Tandem Repeat type of galectins. Immunoblotting showed that neither galectin-2 nor galectin-4 interacted with the components of the mTORC1 machinery that we tested, except for ATP6V1B2, which interacted with galectin-4 (Fig. 1h). Collectively, these observations indicate that galectin-3 interacts with Rag GTPases and Ragulator of the mTORC1 machinery on the lysosomal surface.

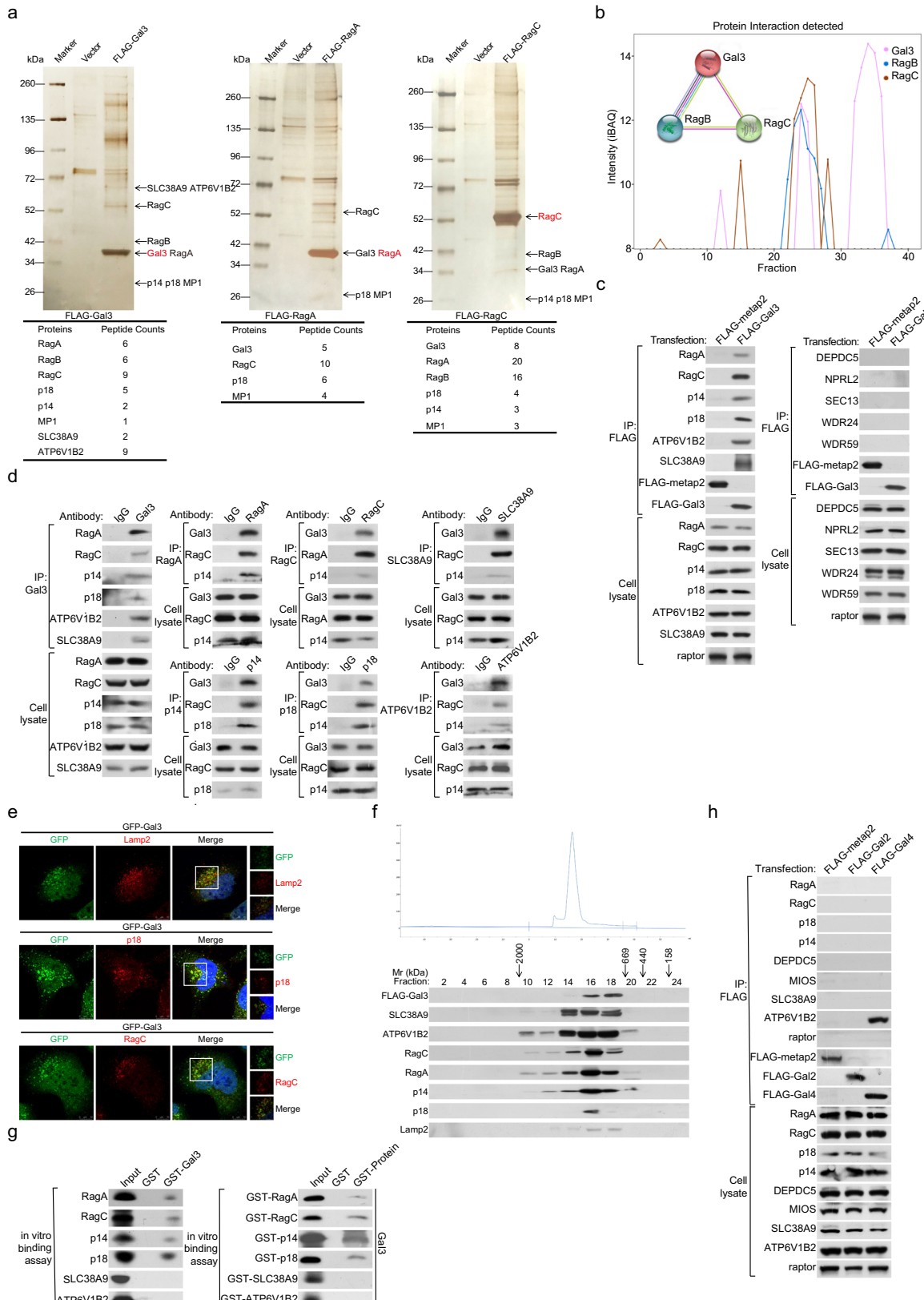

## Galectin-3 facilitates the interaction of Rag GTPases and Ragulator to activate mTORC1

To explore the functional significance of the physical interaction of galectin-3 with Rag GTPases and Ragulator of the mTORC1 machinery, we first investigated the stoichiometry involved in the interaction. Co-immunoprecipitation experiments in HEK-293T cells showed that,

compared to the expression of RagA/RagC alone, co-expression of RagA/RagC together with p14/p18 rendered more efficient interaction of galectin-3 with Rag GTPases and Ragulator (Fig. 2a, Supplementary Fig. 1a). Interestingly, co-expression of RagA/RagC and p14/p18 together with galectin-3 showed a strongly enhanced interaction between RagA/RagC and p14/p18 (Fig. 2a, Supplementary Fig. 1a). In agreement,

**Fig. 1 | Galectin-3 interacts with the components of the mTORC1 signaling machinery on lysosomal surface. a** Immunopurification and mass spectrometry analysis of Gal3-, RagA-, or RagC-associated proteins in HEK-293T cells. Cellular extracts from HEK-293T cells expressing FLAG-Gal3, FLAG-RagA, or FLAG-RagC were subjected to affinity purification with an anti-FLAG affinity column and eluted with excess FLAG peptides. The eluates were resolved by SDS-PAGE and silver-stained. The protein bands were retrieved and analyzed by mass spectrometry. **b** Galectin-3 is co-immunoprecipitated with RagB and RagC. The schematics were from the BioPlex, STRING, and CF-MS explorer databases. **c** HEK-293T cells were transfected with FLAG-Gal3 or FLAG-metap2. Cellular lysates were immunoprecipitated with anti-FLAG followed by immunoblotting with antibodies against the indicated proteins. Each experiment was repeated three times with similar results. **d** Co-immunoprecipitation in HEK-293T cells with anti-galectin-3 followed by immunoblotting with antibodies against the indicated proteins, or immunoprecipitation with antibodies against the indicated proteins followed by immunoblotting with antibodies against galectin-3 or against the components of the

mTORC1 signaling. Each experiment was repeated three times with similar results. **e** HEK-293T cells were transfected with GFP-Gal3 followed by immunofluorescent staining for Lamp2 (red), p18 (red), or RagC (red). Scale bar: 10 μm. Each experiment was repeated three times with similar results. **f** FPLC analysis of FLAG affinity eluates in HEK-293T cells stably expressing FLAG-Gal3. Chromatographic elution profiles and immunoblotting analysis of the chromatographic fractions are shown. Equal volume from each fraction was analyzed, and the elution position of calibration proteins with known molecular masses (kDa) are indicated. Western blotting of galectin-3-containing complex fractionated by Superose 6 gel filtration is shown. **g** GST pull-down assays with GST-fused galectin-3, RagA, RagC, p14, p18, SLC38A9, or ATP6V1B2 and in vitro transcribed/translated proteins as indicated. Each experiment was repeated three times with similar results. **h** Co-immunoprecipitation in HEK-293T cells transfected with FLAG-Gal2 or FLAG-Gal4 with anti-FLAG followed by immunoblotting with antibodies against the indicated proteins. Each experiment was repeated three times with similar results.

the interaction between Rag GTPases and Ragulator was overtly weakened when galectin-3 was stably depleted via lentivirally delivered shRNA in HEK-293T cells (Fig. 2b). These results suggest a role for galectin-3 in facilitating the interaction of Rag GTPases and Ragulator in the mTORC1 signaling.

To further investigate the biological significance of the physical interaction of galectin-3 with Rag GTPases and Ragulator of the mTORC1 machinery, galectin-3 was stably depleted in HEK-293T cell lines. We found that galectin-3 deficiency was associated with a significant decrease in the mTOR kinase activity; the phosphorylation level of S6K1, a downstream target of mTORC1, was significantly reduced (Fig. 2c). Consistently, in mouse embryonic fibroblasts (MEFs), knockdown of galectin-3 by small interfering RNA (siRNA) led to a significantly reduced phosphorylation of not only S6K1, but also 4E-BP1, another mTORC1 substrate (Fig. 2d, Supplementary Fig. 1b). On the contrary, overexpression of galectin-3 in HEK-293T cells resulted in a robust augment of S6K1 phosphorylation (Fig. 2c). However, when the mTOR inhibitor Torin1 was administered to HEK-293T cells overexpressing galectin-3, the phosphorylation of S6K1 was completely inhibited (Fig. 2c), a strong indication that galectin-3 is involved in the mTORC1 signaling. In support of this notion, FLAG-S6K1 and HA-Gal3 were co-overexpressed in galectin-3-deficient HEK-293T cells. Western blotting showed that ectopic expression of galectin-3 led to an enhanced S6K1 phosphorylation in a galectin-3 dose-dependent fashion (Fig. 2e). Importantly, when HEK-293T cells and MEFs were treated with GB1107, a specific membrane-penetrating inhibitor of galectin-3[50,51], the phosphorylation of S6K1 and 4E-BP1 was reduced in a GB1107 dose-dependent manner (Fig. 2f). No evident change in S6K1 phosphorylation was detected when galectin-2 or galectin-4 was overexpressed or knocked down in HEK-293T cells (Fig. 2g). Together, these results indicate that galectin-3 specifically activates the mTORC1 signaling pathway.

**Galectin-3 senses LPS to activate the mTORC1 signaling**
To gain further insights into the role of galectin-3 in the mTORC1 signaling, we next investigated the cellular environments involved in galectin-3-regulated mTORC1 activation. It is reported that amino-acid starvation inhibits the targeting of mTOR to lysosomes thus attenuates the mTORC1 signaling, which can be restored by replenishing amino acids[24,32]. On this ground, we first tested whether amino acids could influence galectin-3-regulated mTORC1 signaling. As stated above, overexpression of galectin-3 resulted in activation of the mTORC1 signaling, whereas knockdown of galectin-3 led to attenuation of the mTORC1 signaling in HEK-293T cells. However, neither deprivation nor supplementation of a panel of amino acids, including leucine and arginine, affected galectin-3-regulated mTORC1 activation (Fig. 3a). We also tested a series of biomolecules, including glucose, lactose, and galactose, and found

that galectin-3-regulated mTORC1 signaling was not influenced by these molecules.

As mentioned earlier, galectin-3 is a member of the β-galactoside binding protein family. It contains in C-terminal a special carbohydrate recognition domain (CRD) that recognizes glycan-containing β-galactoside residues[9,52,53]. Prompted by this, we deprived HEK-293T cells of glucose and amino acids, and found that the activity of mTORC1 was significantly attenuated under this condition (Fig. 3b). Strikingly, treatment of glucose/amino acids-deprived HEK-293T cells with *E. coli* LPS (O111:B4), a complex of lipids and polysaccharides containing β-galactoside residues, led to a significant elevation of the mTORC1 activity (Fig. 3b). However, the LPS-stimulated mTORC1 activity was abolished by not only the mTOR inhibitor Torin1, but also the galectin-3 inhibitor GB1107 (Fig. 3b). Moreover, when HEK-293T cells with similar a galectin-3 expression level were treated with LPS, phosphorylation of S6K1 increased in LPS dose-dependent manner (Supplementary Fig. 2a). Consistently, knockdown of galectin-3 in HEK-293T cells abrogated LPS-stimulated mTORC1 activity, whereas overexpression of galectin-3 in HEK-293T cells augmented LPS-stimulated mTORC1 activation, an effect that was abolished by either GB1107 or Torin1 (Fig. 3c). Similar results were also obtained in MEFs (Fig. 2d, Supplementary Fig. 1b). Notably, both the "atypical" N-terminal domain and the C-terminal carbohydrate recognition domain of galectin-3 were required for sensing LPS to activate mTORC1; absence of either part diminished the LPS-induced mTOR activation (Supplementary Fig. 2b). Moreover, consistent with the earlier observations, overexpression or knockdown of neither galectin-2 nor galectin-4 in HEK-293T cells affected LPS-stimulated mTORC1 activity (Fig. 2g). We also performed epistasis experiments with established mTORC1 regulators to position the galectin-3 function within the mTORC1 pathway. Overexpression of galectin-3 enhanced LPS-stimulated mTORC1 signaling, but not when Rag heterodimer was constitutively devitalized RagA$^{GDP}$/RagC$^{GTP}$ (Fig. 3d), suggesting that LPS-galectin-3 functions upstream of Rag GTPases and highlighting the importance of the interaction of galectin-3 with Rag GTPases and Ragulator in its role in activating mTORC1. Together, these observations support a notion that intracellular galectin-3 senses LPS to stimulate the mTORC1 activity.

**Galectin-3 is a sensor of LPS to promote the activation of Rag GTPases and the targeting of mTOR to lysosomal surface**
To gain further support of the functional role of galectin-3 in sensing LPS and in regulating the mTORC1 signaling, immunofluorescent staining in HEK-293T cells found that both LPS and galectin-3 were co-localized with the lysosomal marker Lamp2 (Fig. 4a). Importantly, peptide pull-down experiments with biotinylated LPS and bacterially-purified galectin-3, p14/p18, or RagA/RagC showed that LPS exhibited indeed a high affinity toward only galectin-3 (Fig. 4b). Furthermore,

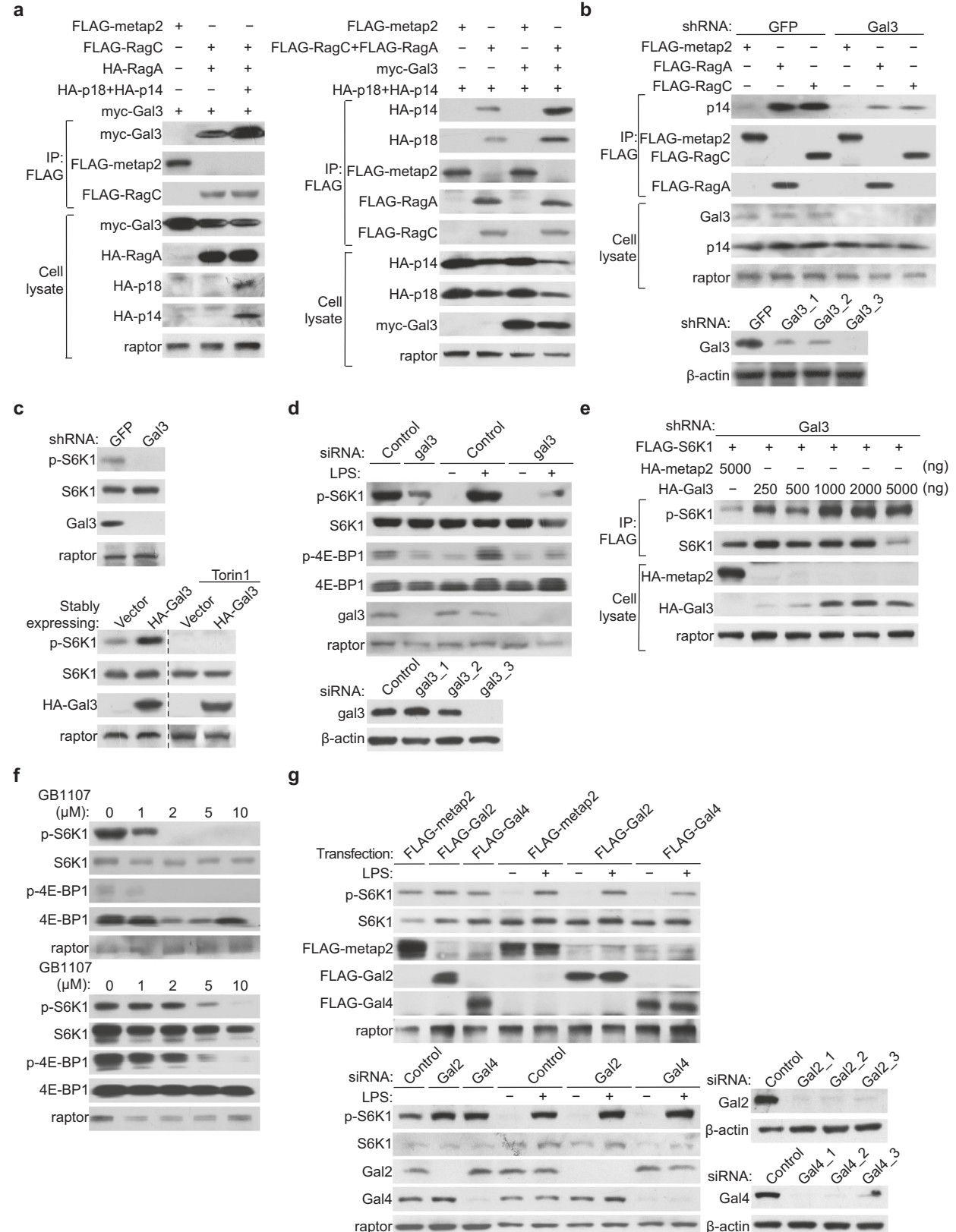

computational analysis based on the structure of galectin-3 and LPS from the Protein Data Bank (3zsj and 6s8h) revealed that the CRD domain of galectin-3 is capable of binding the polysaccharide of LPS[54,55], and that this binding would result in a conformational change of galectin-3 that might drive the interaction of galectin-3 with Rag GTPases and Ragulator (Fig. 4c). In addition, there was early

evidence that intracellular LPS can bind to galectin-3[9]. Indeed, co-immunoprecipitation experiments in HEK-293T cells showed that in the presence of LPS, the interaction of galectin-3 with Ragulator (p14/p18) especially with RagA/RagC was greatly enhanced, while no inter-action of galectin-3 with GATOR1 (DEPDC5/NPRL2) or GATOR2 (SEC13, WDR24, and WDR59) was detected (Fig. 4d), consistent with the earlier

**Fig. 2 | Galectin-3 facilitates the interaction of Rag GTPases and Ragulator to stimulate the mTORC1 signaling. a** HEK-293T cells were transfected with the indicated plasmids for co-immunoprecipitation with anti-FLAG followed by immunoblotting with antibodies against the indicated proteins. **b** HEK-293T cells were infected with lentiviruses carrying shGFP or shGal3 and transfected with FLAG-metap2, FLAG-RagA, or FLAG-RagC. Cellular lysates were immunoprecipitated with anti-FLAG followed by immunoblotting with antibodies against the indicated proteins. Each experiment was repeated three times with similar results. **c** HEK-293T cells were infected with lentiviruses carrying shGFP or shGal3, or HA-Vector or HA-Gal3 with or without the treatment with the mTOR inhibitor Torin1 (250 nM) for western blotting analysis of the level or phosphorylation of the indicated proteins. Each experiment was repeated three times with similar results. **d** MEFs cells were treated with galectin-3 siRNA, starved of amino acids and glucose

for 1 h, and stimulated with LPS (1 μg/ml) for 5 h for western blotting analysis of the level or phosphorylation of the indicated proteins. **e** Galectin-3-deficient HEK-293T cells were co-transfected with FLAG-S6K1 and HA-metap2 or increasing amounts of HA-Gal3. Cellular lysates were immunoprecipitated with anti-FLAG followed by immunoblotting with antibodies against the indicated proteins. Each experiment was repeated three times with similar results. **f** HEK-293T (upper) and MEFs (lower) were treated with different concentrations of GB1107 for western blotting analysis of the level or phosphorylation of the indicated proteins. Each experiment was repeated three times with similar results. **g** HEK-293T cells were treated with siRNAs against galectin-2 or galectin-4 or transfected with FLAG-Gal2 or FLAG-Gal4, starved of amino acids and glucose for 1 h, and stimulated with LPS (1 μg/ml) for 5 h for western blotting analysis of the level or phosphorylation of the indicated proteins. Each experiment was repeated three times with similar results.

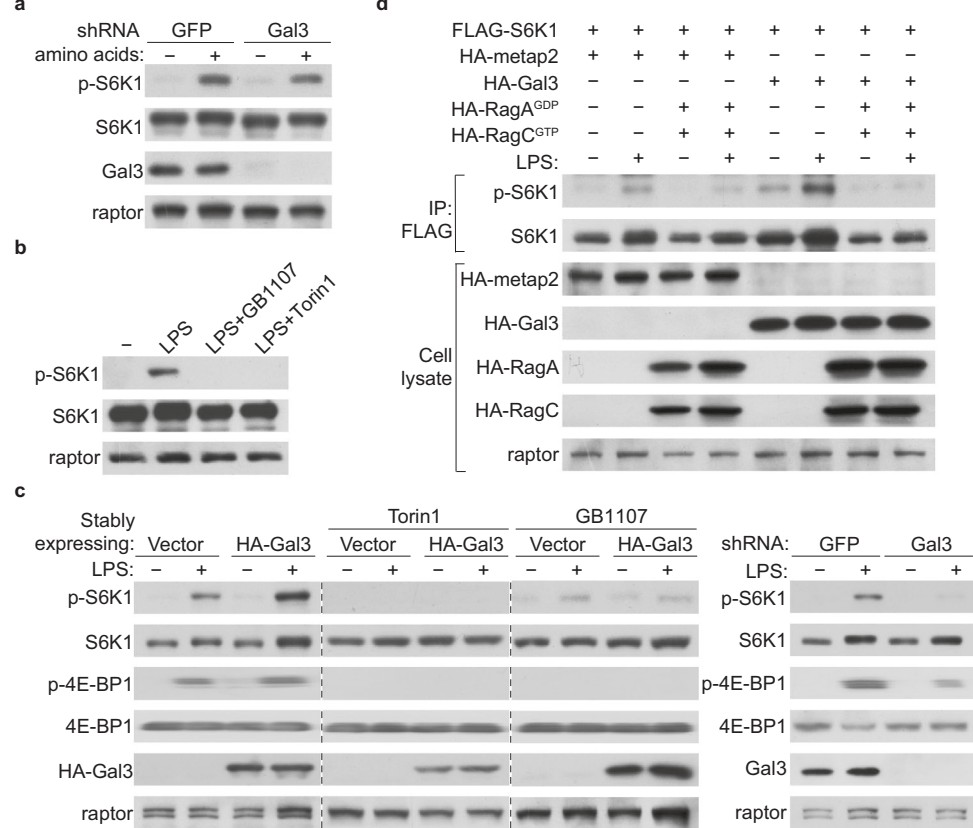

**Fig. 3 | Galectin-3 senses LPS to activate mTORC1. a** HEK-293T cells were infected with lentiviruses carrying shGFP or shGal3, starved of amino acids for 50 min, and replenished with amino acids for 10 min for western blotting analysis of the level or phosphorylation of the indicated proteins. Each experiment was repeated three times with similar results. **b** HEK-293T cells were starved of amino acids and glucose for 1 h followed by stimulation with LPS (1 μg/ml) for 5 h in the presence or absence of the mTOR inhibitor Torin1 (250 nM) or galectin-3 inhibitor GB1107 (5 μM) for western blotting analysis of the level or phosphorylation of the indicated proteins. Each experiment was repeated three times with similar results. **c** HEK-293T cells were infected with lentiviruses carrying shGFP or shGal3, or HA-Vector or HA-Gal3

treated with or without GB1107 (5 μM) or Torin1 (250 nM), and starved of amino acids and glucose for 1 h followed by stimulation with LPS (1 μg/ml) for 5 h for western blotting analysis of the level or phosphorylation of the indicated proteins. Each experiment was repeated three times with similar results. **d** HEK-293T cells were transfected with the indicated plasmids, starved of amino acids and glucose for 1 h, and stimulated with LPS (1 μg/ml) for 5 h. Cellular lysates were immunoprecipitated with anti-FLAG followed by immunoblotting with antibodies against the indicated proteins. Each experiment was repeated three times with similar results.

observations. These results support the proposition that galectin-3 is a sensor of LPS, and these observations also support a notion that the binding of LPS to galectin-3 promotes the interaction of galectin-3 with Rag GTPases and Ragulator.

The transition of the nucleotide state of Rag GTPases is the key in the activation of the mTORC1 pathway, which is regulated by the interaction of Rag GTPases with Ragulator[33,56]. To gain further support of LPS-promoted interaction of galectin-3 with Rag GTPases and Ragulator and to explore the biological significance involved in the

interaction, we next investigated the relationship between the interaction of galectin-3 with RagA/RagC and Ragulator and the nucleotide state of Rag GTPases. To this end, we constructed two classes of Rag mutants with different nucleotide binding states, one class was dominant negative forms, RagA[T21N] and RagC[S75N], that cannot bind GTP, and the other class included RagA[Q66L] and RagC[Q120L] that constitutively bind GTP[30,57] (Fig. 4e). Co-expression of wild-type RagA/RagC or mutated RagA/RagC together with p14/p18 and galectin-3 in HEK-293T cells followed by co-immunoprecipitation in the presence of LPS

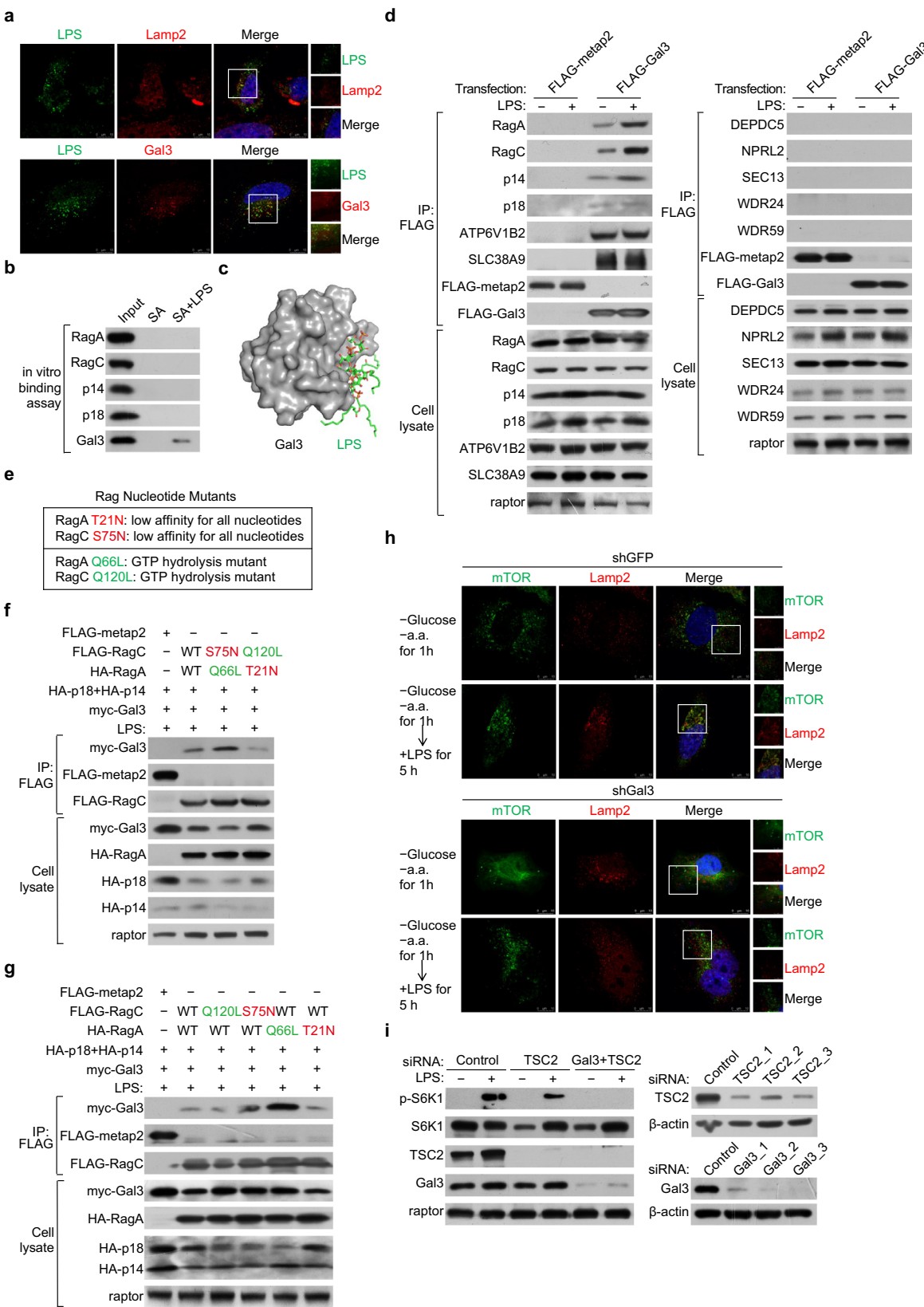

showed that galectin-3 interacted with RagA$^{Q66L}$/RagC$^{S75N}$ much stronger than with RagA$^{T21N}$/RagC$^{Q120L}$ (Fig. 4f), suggesting that galectin-3 is preferably associated with activated Rag GTPases (RagA$^{Q66L}$/RagC$^{S75N}$) to inactive Rag GTPases (RagA$^{T21N}$/RagC$^{Q120L}$). Moreover, co-immunoprecipitation experiments in HEK-293T cells co-expressing galectin-3 and mutant RagA or galectin-3 and mutant RagC showed

that the heterodimer composed of GTP-bound RagA and wild-type RagC (RagA$^{Q66L}$/RagC) interacted with galectin-3 more strongly than the heterodimers composed of either wild-type RagA and GTP-bound RagC (RagA/RagC$^{Q120L}$), wild-type RagA and nucleotide-free RagC (RagA/RagC$^{S75N}$), or nucleotide-free RagA and wild-type RagC (RagA$^{T21N}$/RagC) (Fig. 4g), suggesting that the interaction of galectin-3

**Fig. 4 | Galectin-3 binds LPS to promote the activation of Rag GTPases and the targeting of mTOR to lysosomal surface. a** HEK-293T cells were stimulated with LPS (1 μg/ml) for 6 h followed by immunofluorescent staining for LPS (green) and Lamp2 (red) or galectin-3 (red). Scale bar: 10 μm. Each experiment was repeated three times with similar results. **b** In vitro purified RagA or RagC, p14 or p18, or galectin-3 was incubated with biotinylated LPS and mixed with streptavidin beads. The bound proteins were eluted for immunoblotting analysis with antibodies against the indicated proteins. Each experiment was repeated three times with similar results. **c** Prediction of the binding pose of LPS to galectin-3. **d** HEK-293T cells were transfected with FLAG-Gal3 or FLAG-metap2, starved of amino acids and glucose for 1 h, and stimulated with LPS (1 μg/ml) for 5 h. Cellular lysates were immunoprecipitated with anti-FLAG followed by immunoblotting with antibodies against the indicated proteins. Each experiment was repeated three times with similar results. **e** The Rag GTPases mutants with different nucleotide states. **f** HEK-293T cells were co-transfected with the indicated plasmids and treated with LPS (1 μg/ml) for 6 h. Cellular lysates were immunoprecipitated with anti-FLAG followed by immunoblotting with antibodies against the indicated proteins. Each experiment was repeated three times with similar results. **g** HEK-293T cells were co-transfected with the indicated plasmids and RagA or RagC with different nucleotide states treated with LPS (1 μg/ml) for 6 h. Cellular lysates were immunoprecipitated with anti-FLAG followed by immunoblotting with antibodies against the indicated proteins. Each experiment was repeated three times with similar results. **h** HEK-293T cells were infected with lentiviruses carrying shGFP or shGal3, starved of amino acids and glucose for 1 h, and stimulated with LPS (1 μg/ml) for 5 h followed by immunofluorescent staining for mTOR (green) and Lamp2 (red). Scale bar: 10 μm. Each experiment was repeated three times with similar results. **i** HEK-293T cells were treated with siRNAs targeting TSC2 or TSC2 plus galectin-3, starved of amino acid and glucose for 1 h, and stimulated with LPS (1 μg/ml) for 5 h for western blotting analysis of the level or phosphorylation of the indicated proteins. Each experiment was repeated three times with similar results.

with RagA/RagC is functionally linked to the nucleotide state of RagA and the generation of active Rag GTPases (RagA$^{GTP}$/RagC$^{GDP}$). Together, these observations support a notion that galectin-3 senses and binds LPS, leading to an enhanced interaction of galectin-3 with Rag GTPases and Ragulator, thereby activating Rag GTPases and the mTORC1 signaling.

Another key step in the activation of mTORC1 signaling is the targeting of mTOR to lysosomal surface[33,56]. To gain further support for the proposition that galectin-3 senses LPS to stimulate the mTORC1 activity, immunofluorescent staining in HEK-293T cells showed that deprivation of amino acids and glucose negatively regulated the targeting of mTOR to Lamp2-positive lysosomes (Fig. 4h). Remarkably, the targeting of mTOR to Lamp2-positive lysosomes was restored by LPS stimulation, an effect that was abolished by knockdown of galectin-3 (Fig. 4h).

It has also been reported that LPS regulates mTORC1 through the PI3K-Akt-TSC-mTORC1 signaling pathway[58–60]. We thus knocked down TSC2 in HEK-293T cells to disrupt this pathway. Western blotting indicated that HEK-293T cells deficient of TSC2 still respond to LPS to activate mTORC1 (Fig. 4i). However, when galectin-3 was co-knocked down, LPS was no longer capable of activating mTORC1 (Fig. 4i), arguing against the possibility that LPS/galectin-3 regulates mTORC1 through the PI3K-Akt-TSC-mTORC1 signaling pathway.

## Galectin-3 promotes glycolysis through activating the mTORC1 signaling

Galectin-3 (*LGALS3*) appears to emerge late in evolution; its orthologs are found in vertebrates including *Xenopus tropicalis* but not in other established model organisms including *Saccharomyces cerevisiae* (Fig. 5a). Potential homologues are also found in higher invertebrates such as *Acanthaster planci* and *Aplysia californica* (Fig. 5a). To further support LPS/galectin-3-activated mTORC1 signaling and to explore the biological significance of this activation, we next investigated the transcriptomic profile associated with LPS/galectin-3-activated mTORC1. To this end, we first analyzed the differentially expressed genes in HEK-293T cells with or without the treatment of siGal3 or Torin1 by RNA-based deep sequencing (RNA-seq). Total RNAs were extracted for cDNA synthesis, library construction, and sequencing using BGI-seq 500 (LC-Bio, Hangzhou) with a signal end of 50-bp reads. The quality of raw data was monitored by FastQC, and low-quality reads and sequencing adapters including those with more than 5 'N' bases and an average phred quality score less than 15 were removed by fastp. With *P* < 0.05 and absolute log 2 (fold change) > 1, 630 differentially expressed genes were extracted in siGal-3 group, of which 334 were up-regulated and 296 were down-regulated, while 2,713 differentially expressed genes were extracted in Torin1 group, with 1,032 up-regulated and 1681 down-regulated (GSE211784 [https://www.ncbi.nlm.nih.gov/geo/query/acc.cgi?acc=GSE211784]) (Fig. 5b). Cross

analysis between these two groups yielded 25 down-regulated genes, including those that are implicated in several important cellular signaling pathways such as cell growth and metabolism (Fig. 5c). Among the down-regulated genes upon galectin-3 knockdown or mTOR inhibition, *HK2* and *PKM* encode for important enzymes functioning in glycolysis; HK2 catalyzes the first step of glycolysis, while PKM catalyzes the last step of glycolysis[61]. In addition, *GLUT1*, a gene functionally associated with glucose uptake and glycolysis[62,63], was also among the down-regulated genes upon galectin-3 knockdown or mTOR1 inhibition (Fig. 5c).

To validate the RNA-seq results, real-time reverse transcriptase PCR (qPCR) was performed and the expression of the representative genes, including *HK2*, *PKM2*, and *GLUT1*, as well as other genes that are implicated in glycolysis including glucose-6-phosphate isomerase (*GPI*), phosphofructokinase (*PFKM*), and lactate dehydrogenase A (*LDHA*), were analyzed in HEK-293T cells under galectin-3 knockdown or Torin1 treatment. The results showed that knockdown of galectin-3 or treatment with Torin1 was associated with a significant decrease in the expression of *HK2*, *PKM2*, and *GLUT1*, whereas the expression of *GPI*, *PFKM*, and *LDHA* was unaffected (Fig. 5d). Meanwhile, western blotting analysis in HEK-293T cells showed that knockdown of galectin-3 resulted in a decreased whereas overexpression of galectin-3 led to an increased expression of HK2, PKM2, and GLUT1 (Fig. 5e). Importantly, consistent with our working model, the addition of LPS in HEK-293T cells was associated with a marked increase in the protein expression of HK2, PKM2, and GLUT1, an effect that was abolished by GB1107 or Torin1 (Fig. 5f). Inhibition of mTORC1 activity has been linked to autophagy initiation, leading to activation of the degradation pathway[64,65]. To investigate the effect of autophagy on the expression of HK2, PKM2 and GLUT1, we simultaneously knocked down galectin-3 and ULK1 (unc-51-like autophagy activating kinase 1) in HEK-293T cells, and found that autophagy inhibition could not restore the expression of HK2, PKM2 and GLUT1 (Supplementary Fig. 3a).

As stated before, it is proposed that the galectin-3 is beneficial to glucose homeostasis and has a protective effect on diabetogenesis when nutrients are excess[18,19]. Based on our observations that galectin-3 regulates the expression of genes that are critically involved in glycolysis, it is possible that high level of galectin-3 is associated with an increased glucose uptake and glycolysis, thereby regulating blood level of glucose in diabetogenesis. To test this hypothesis, FLAG-gal3 was expressed in HEK-293T cells deficient of galectin-3. Biochemical analysis in these cells indicated that overexpression of galectin-3 was associated with a significant increase in glucose consumption and lactate production, which was greatly augmented by LPS (Fig. 5g). However, the increased glucose consumption and lactate production were abrogated when GB1107 or Torin1 was added to the cells (Fig. 5g). To exclude the possibility that of Torin1 effects mTORC2, knockdown raptor also resulted in a reduced glucose consumption and lactate production (Supplementary Fig. 3b).

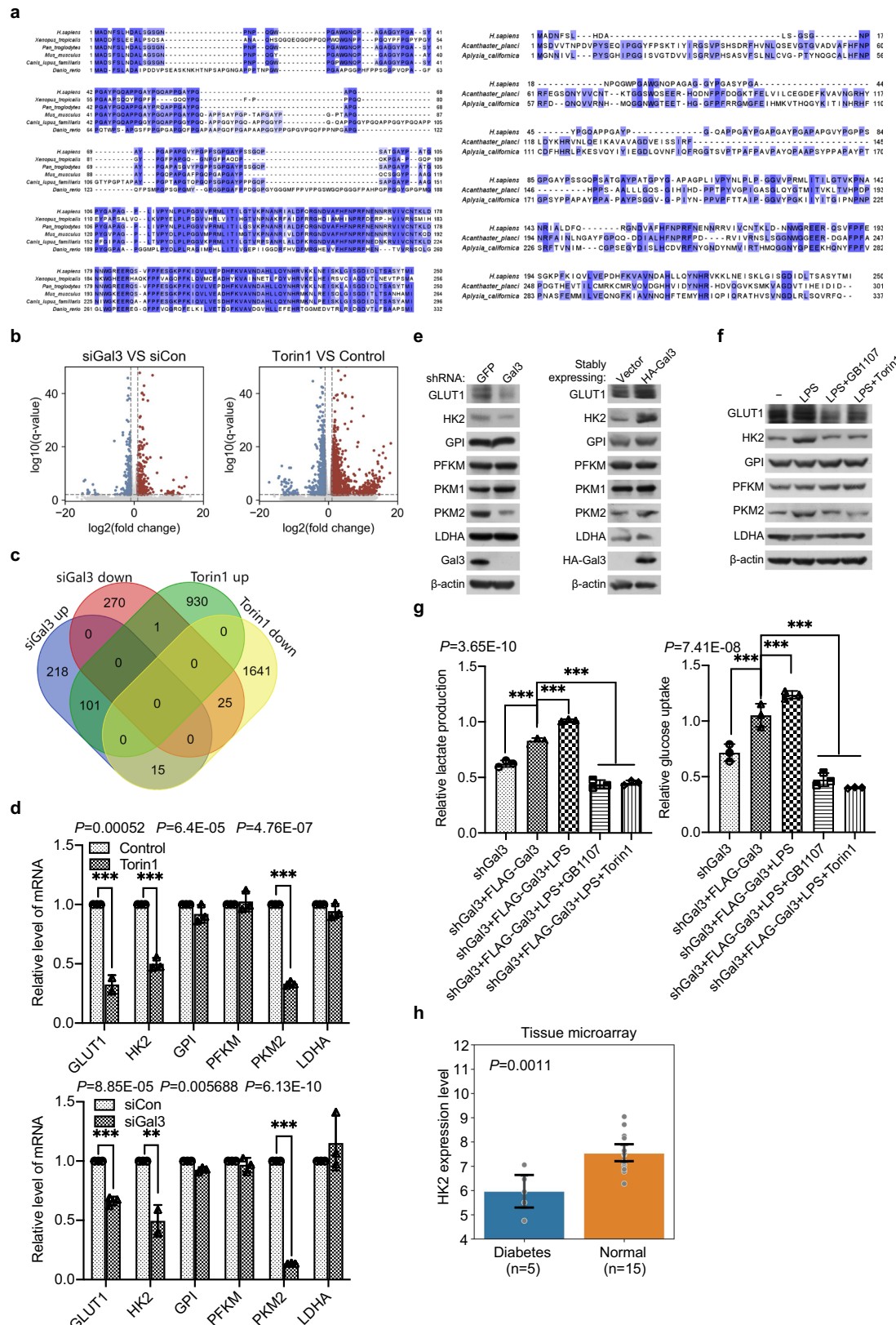

To extend our observations to clinicopathologically relevant settings, computational analysis of diabetes patient data from ArrayExpress (E-MEXP-2559) found that the expression level of the key glycolytic enzyme HK2 was significantly reduced in skeletal muscles of diabetic patients (Fig. 5h). Due to a lack of appropriate diabetes datasets, further analysis of the LPS/galectin-3-Rag GTPases/Ragulator-mTORC1 axis in the context of diabetes was not possible in the current study. Nevertheless, given the proposition that high galectin-3 has a protective effect on obesity and diabetogenesis under overnutrition[15,18,19], our results support a notion that the LPS/galectin-3-mTOR1 signaling pathway promotes glucose uptake and glycolysis through regulating the expression of the glycolytic genes including *GLUT1*, *HK2*, and

**Fig. 5 | Galectin-3 promotes glycolysis through activating the mTORC1 signaling. a** *LGALS3* homologues in vertebrates and invertebrates. Amino-acid positions were colored in white to blue. **b** HEK-293T cells were treated with siGal3 or Torin1 (250 nM). Total RNAs were extracted for RNA-seq. The volcano plots delineate up- and down-regulated genes analyzed with $P < 0.05$ and absolute log 2 (fold change) > 1. $P$ values attained by the two-sided Wald test are corrected for multiple testing using the Benjamini and Hochberg method, and shown as $q$ values. **c** The Venn diagram depicts the genes cross-analyzed from the two sets of RNA-seq experiments. **d** qPCR measurement of the expression of the indicated genes selected from RNA-seq results in HEK-293T cells treated with siGal3 or Torin1 (250 nM). Each bar represents the mean ± SD for biological triplicate experiments. $P$ values were calculated by two-tailed unpaired Student's $t$ test (**$P < 0.01$, ***$P < 0.001$). **e** HEK-293T cells were infected with lentiviruses carrying shGFP or shGal3, or HA-Vector or HA-Gal3 for western blotting analysis of the level of the indicated proteins. Each experiment was repeated three times with similar results. **f** HEK-293T cells were starved of amino acids and glucose for 1 h followed by stimulation with LPS (1 μg/ml) for 5 h in the presence or absence of Torin1 (250 nM) or GB1107 (5 μM) for western blotting analysis of the level of the indicated proteins. **g** HEK-293T cells deficient of galectin-3 were transfected with FLAG-Gal3 and stimulated with LPS in the presence or absence of GB1107 or Torin1 for the measurement of glucose uptake and lactate production. Each bar represents the mean ± SD for biological triplicate experiments. $P$ values were calculated by one-way ANOVA (***$P < 0.001$). **h** Bioinformatics analysis of the public datasets of diabetes patient skeletal muscle samples and normal tissues. $P$ value was calculated by two-tailed unpaired Student's $t$ test (**$P < 0.01$). Error bars represent 95% confidence intervals.

*PKM2*, thereby functionally contributing to glucose homeostasis and diabetogenesis.

## Galectin-3 is overexpressed in HCC and implicated in hepatocarcinogenesis

Metabolic dysregulation is a hallmark of cancerous cells[66,67]. Unlike normal cells, which derive most of their energy from mitochondrial oxidative phosphorylation, cancer cells heavily rely on glycolysis as their primary energy resource, a phenomenon known as the "Warburg effect"[68,69]. Normally, the expression of galectin-3 is low in most of the human tissues, especially in the liver (Fig. 6a). However, analysis of the public datasets in Oncomine (https://www.oncomine.org/) showed that galectin-3 is highly expressed in a number of malignancies; galectin-3 mRNA level is significantly upregulated in thyroid gland papillary carcinoma, prostatic intraepithelial neoplasia, and especially in HCC (Fig. 6b). To corroborate this, we compared the expression of galectin-3 in HCC samples versus the corresponding adjacent samples in TCGA, and found that the level of galectin-3 expression in HCC samples was significantly higher than that in adjacent samples (Fig. 6c). Analysis the public database in the Human Pathology Atlas also indicated that galectin-3 is highly expressed in HCC (Fig. 6d).

Deregulation of cellular energetics involving an increase in glycolysis is an eminent feature of hepatocarcinogenesis[10,50]. Based on the observations that galectin-3 is overexpressed in HCC, and that galectin-3 activates the mTORC1 signaling to regulate the expression of genes that are critically involved in glycolysis, it is reasonable to postulate that galectin-3 promotes glucose uptake and glycolysis through regulating the mTORC1 signaling pathway, leading to the development and/or progression of HCC. To test this, we first performed co-immunoprecipitation assays in HepG2 cells and confirmed the interaction of galectin-3 with Rag GTPases (RagA/RagC) and Ragulator (p14/p18) in HCC (Fig. 6e). We then investigated the effect of galectin-3 on mTORC1 activity in HepG2 cells and found that knockdown of galectin-3 was associated with an inhibition of the mTORC1 activity, whereas overexpression of galectin-3 led to an elevation of the mTORC1 activity (Fig. 6f). Moreover, silencing of the expression of galectin-3 in HepG2 cells resulted in a marked decrease in the expression of HK2, PKM2, and GLUT1 (Fig. 6g).

The liver is an important place for LPS removal, and the concentration of LPS in liver is high[70]. Immunoblotting experiments showed that LPS treatment of HepG2 cells led to a significant increase in the phosphorylation level of S6K1, as well as an increase in the expression of HK2, PKM2, and GLUT1 (Fig. 6h). Consistent with the earlier observations, LPS-stimulated phosphorylation of S6K1 and expression of HK2, PKM2, and GLUT1 were abrogated by either Torin1 or GB1107 (Fig. 6h). In agreement, LPS treatment of HepG2 cells was associated with a significant increase in glucose uptake and lactate production, effects that were abolished by knockdown of galectin 3 in the cells or treatment of the cells with GB1107 or Torin1 (Fig. 6i). Together, these observations support the notion that LPS/galectin-3 promotes glycolysis in HCC through activating the mTORC1 signaling.

To support the proposition that LPS/galectin-3 is involved in the development and/or progression of HCC. Cell proliferation assays using CCK-8 (Cell Counting Kit-8) demonstrated that the addition of LPS promoted the proliferation of HepG2 cells, an effect that was, at least partially, offset in the presence of GB1107 (Fig. 7a). Meanwhile, knockdown of galectin-3 was associated with an inhibition of the proliferation of HepG2 cells, an effect that was rescued by overexpression of RagA$^{GTP}$/RagC$^{GDP}$ (Fig. 7a). In addition, colony formation experiments showed that LPS promoted HepG2 colony formation, which was largely abrogated by addition of Torin1 or GB1107 (Fig. 7b, Supplementary Fig. 4). Similarly, knockdown of galectin-3 in HepG2 cells led to an inhibited colony formation, which was, at least partially, rescued by overexpression of RagA$^{GTP}$/RagC$^{GDP}$ (Fig. 7b, Supplementary Fig. 4). Together, these results support the notion that LPS/galectin promotes the development and/or progression of HCC through stimulating the mTORC1 signaling.

Finally, to further extend our observations to clinicopathologically relevant settings, Kaplan-Meier survival analysis of the Human Pathology Atlas database (https://www.proteinatlas.org/ENSG00000131981-LGALS3/pathology/liver+cancer) indicated that HCC patients with high expression of galectin-3 experienced a significant unfavorable survival (Fig. 7c). We tried all possible cutoffs, and the best cutoff divided patients into two groups with more pronounced differences in prognosis (Fig. 7c). We also analyzed the transcriptomics datasets from TCGA primary tumor samples (10,496 cases in total) and GTEX normal tissue samples (7,792 cases in total) from UCSD Xena, and found that the expression level of galectin-3 was correlated with that of HK2 and PKM in all data (Fig. 7d). In addition, analysis of the transcriptomics data and matched survival data of TCGA HCC patients from UCSD Xena with the median expressions of galectin-3, HK2, and PKM as cutoffs revealed that the survival of the patients with a combined high expression of galectin-3 and HK2 or galectin-3 and PKM was significantly worse than the patients with a combined low expression of galectin-3 and HK2 or galectin-3 and PKM (Fig. 7e). These results support a role for galectin-3 in the development and progression of HCC.

## Discussion

Galectin-3 is a member of the evolutionary conserved animal lectins family. It is widely expressed in cells and plays important roles in different subcellular compartments. In the past, a few decades, the research on intracellular galectin-3 has mainly focused on apoptosis, autophagy, and inflammation[7–9]. We report in the current study that galectin-3 interacts with the Rag GTPases and Ragulator of the mTORC1 signaling machinery on lysosomes. This finding not only implicates galectin-3 in mTORC1 signaling and cellular metabolism, but also expands the subcellular compartmentalization of galectin-3 to include lysosomes, although it has been reported that galectin-3 forms puncta in lysosomes when the lysosomal membrane is damaged[71–73]. Whether or not galectin-3 also presents and functions on the membranes of other subcellular organelles/compartments needs further investigations, at least on cell membrane, galectin-3 mediates cell

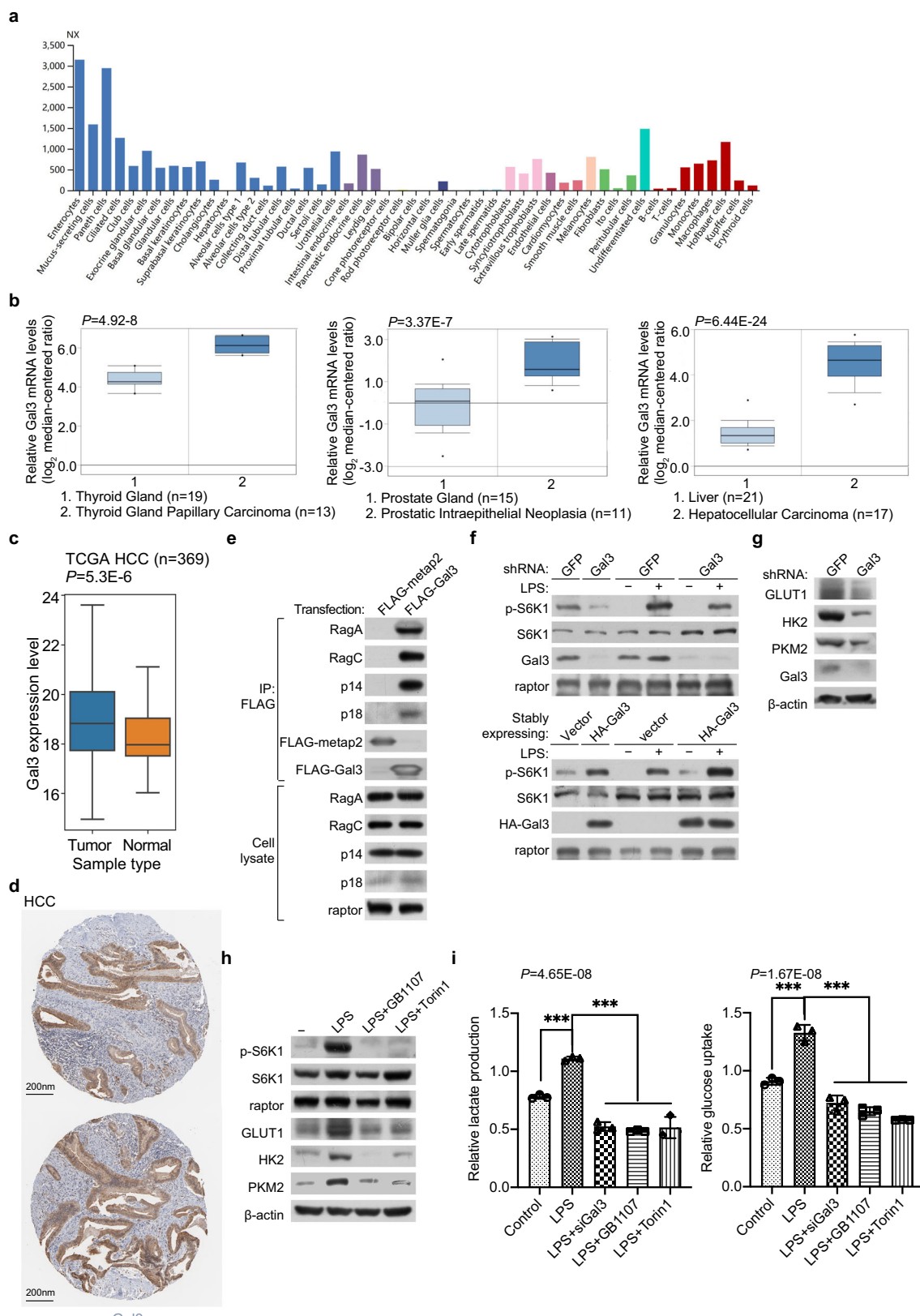

adhesion and cell-cell interaction through specific recognition of different complex carbohydrates on cell surface[4,5], as stated earlier.

In the mTORC1 signaling, lysosome-anchored Ragulator recruits Rag GTPases, leading to the activation of Rag GTPases and subsequent lysosomal loading of mTOR[32]. Based on the results of our study, we propose that galectin-3 acts upstream of Rag GTPases and Ragulator to

facilitate the interaction of these two components and the activation of Rag GTPases. Since galectin-3 does not possess any known enzymatic and biochemical activity, we propose that it functions mainly through its biophysical activity. Protein-protein interaction is an eminent scheme in cell signaling transduction[74], which is often driven by the conformational change of the involving proteins that is imparted or

**Fig. 6 | Galectin-3 is overexpressed in HCC to activate mTORC1 and promote glycolysis. a** Analysis of the expression of galectin-3 in different human tissues. **b** Analysis of the Cancer Genome Atlas datasets in Oncomine for the expression of galectin-3 between tumor and normal tissues. Each bar represents the mean ± SD for biological triplicate experiments. Data are presented with box plots (***$P < 0.001$, by unpaired, two-tailed Student's $t$ test). The minima, maxima, center of the box plots are defined as the 75%, 25% and 50% percentile, the whiskers are defined as the interquartile range times 1.5. **c** Analysis of the expression of galectin-3 in liver cancer samples and corresponding adjacent samples in TCGA HCC. Each bar represents the mean ± SD for biological triplicate experiments. Data are presented with box plots (***$P < 0.001$, two-sided Mann-Whitney $U$ test). The minima, maxima, center of the box plots are defined as the 75%, 25% and 50% percentile, the whiskers are defined as the interquartile range times 1.5. **d** Analysis of the Human Pathology Atlas database for the expression of galectin-3 by immunohistochemistry in HCC samples. **e** HepG2 cells were transfected with FLAG-Gal3 or FLAG-metap2. Cellular lysates were immunoprecipitated with anti-FLAG followed by immunoblotting with antibodies against the indicated proteins. Each experiment was repeated three times with similar results. **f** HepG2 cells were infected with lentiviruses carrying shGFP or shGal3, or HA-Vector or HA-Gal3, starved of amino acids and glucose for 1 h, and stimulated with LPS (1 μg/ml) for 5 h for western blotting analysis of the level or phosphorylation of the indicated proteins. Each experiment was repeated three times with similar results. **g** HepG2 cells were infected with lentiviruses carrying shGFP or shGal3 for western blotting analysis of the level of the indicated proteins. Each experiment was repeated three times with similar results. **h** HepG2 cells were starved of amino acids and glucose for 1 h and stimulated with LPS (1 μg/ml) for 5 h in the presence or absence of Torin1 (250 nM) or GB1107 (5 μM) for western blotting analysis of the level or phosphorylation of the indicated proteins. Each experiment was repeated three times with similar results. **i** HepG2 cells were treated with LPS, LPS and siGal3, LPS and GB1107, or LPS and Torin1 to detect glucose uptake and lactate production. Each bar represents the mean ± SD for biological triplicate experiments. $P$ values were calculated by one-way ANOVA (***$P < 0.001$).

rendered by post-translational modifications, binding with macromolecules, and so on[74,75]. Importantly, we identified in our study that galectin-3 is able to sense and bind LPS. Based on our observations that LPS significantly augments the interaction of galectin-3 with Rag GTPases and Ragulator and enhances galectin-3-stimulated mTORC1 activity, it is conceivable that the binding of LPS by galectin-3 is associated with a change in the conformation of galectin-3 that is conducive for its role in facilitating the interaction of Rag GTPases and Ragulator and the activation of Rag GTPases. Unfortunately, the three-dimensional structure of LPS-bound galectin-3 are unavailable, but our computational analysis based on the docking of LPS and galectin-3 predicted LPS-galectin-3 interaction and suggested the conformational change of galectin-3 associated with LPS binding. Of note, our identification of the interaction of galectin-3 with Rag GTPases and Ragulator by epitope-based proteomic screening with combined immunopurification and mass spectrometry was without addition of exogenous LPS. The existence of endogenous LPS cannot be excluded in our experimental system. Moreover, it is highly possible that other intracellular macromolecules biochemically and structurally resembling LPS could bind to galectin-3, albeit at a lower affinity. Nevertheless, based on the results of our study, we propose that galectin-3 acts as a sensor of LPS to facilitate the interaction between Rag GTPases and Ragulator and the activation of Rag GTPases, stimulating the mTORC1 signaling.

It has been reported that LPS activates mTORC1 through PI3K-Akt-TSC-mTORC1 signaling pathway[58–60]. By knocking down TSC2 to disrupt the PI3K-Akt-TSC-mTORC1 signaling pathway, we found that cells deficient of TSC2 still respond to LPS to activate mTORC1, supporting our working model of a galectin-3 route in the activation of mTORC1. While our study is by no means to exclude the PI3K-Akt-TSC route in activating of mTORC1, the activation of mTORC1 by LPS through two distinct mechanisms is nevertheless intriguing. Future investigations are needed to explore the functional relationship between these two mechanisms and the factors involved to generate a clearer picture about the role of LPS in mTORC1 activation.

It has also been reported that LPS interacts with galectin-3 to regulate noncanonical inflammasome[9]. This is consistent with not only our proposition that galectin-3 is a sensor of LPS, but also our observations that LPS/galectin-3 is involved in the development of diabetes and cancer, both pathological states are intimately associated with inflammatory responses. It has been well documented that inflammation plays a critical role in tumorigenesis[76]. However, a direct causal relationship between inflammation and the development of diabetes is less clear[77]. As stated earlier, LPS is a component of the outer wall of the cytoderm of Gram-negative bacteria[43]. Typically, the blood level of LPS fluctuates with intestinal microbiota, and high level of LPS is associated with chronic subclinical inflammatory processes, obesity, changes in glucose metabolism and even

cancer[44–46]. A plethora of investigations has highlighted the intricate interplay among gut microbiota, dysbiosis, and the development of metabolic syndromes and cancer[48,78,79]. While several studies reported that elevated LPS level results in chronic inflammation leading to the development of diabetes[44,80,81], other studies suggested that elevated LPS level is associated with a reduced incidence of diabetes[47,48]. Clearly, the involvement of LPS in diabetogenesis needs more investigations. Nonetheless, we found in the current study that LPS binds to galectin-3, leading to the activation of mTORC1 and the expression of several key glycolysis-associated genes including *GLUT1*, *HK2*, and *PKM2*, thereby promoting glucose uptake and glycolysis. If our interpretation is correct, our study at least adds an additional line of evidence and a mechanistic insight to support the link between inflammation and diabetes/cancer. In effect, intracellular galectin-3 senses and binds LPS. LPS-bound galectin-3 facilitates the interaction of Rag GTPases and Ragulator, leading to the activation of Rag GTPases and lysosomal targeting of mTOR. This results in the activation of mTORC1, which signals the expression of downstream target genes including *GLUT1*, *HK2*, and *PKM2*. Increased expression of the glycolysis-associated genes leads to an increased glucose uptake and glycolysis, leading to a reliever of blood glucose level. This scenario is consistent with the aforementioned proposition that galectin-3 is beneficial to glucose homeostasis and has a protective effect on diabetogenesis[18,19]. In the case of cancer, increased expression of the glycolysis-associated genes thus increased glucose uptake and glycolysis will feed the growth of cancer cells and promote tumorigenesis. After all, metabolic dysregulation is a hallmark of cancerous cells, which heavily rely on glycolysis as their primary energy resource[66–69], as mentioned above.

It must be pointed out that, due to the complexity of the pathogenesis of both diabetes and cancer, the contribution and position of the LPS/galectin-3-Rag GTPases/Ragulator-mTORC1 axis to diabetogenesis and tumorigenesis need more investigations. Perhaps more relevant to our current study, what is the physiological significance of this axis to cells and to organisms? It is probably a safe assumption that the cell and the life start without the accountment of bacterial LPS. Is the LPS-galectin-3 interaction a result of evolution or adaption? Then, how this interaction was evolved and what it was adapted for? How the good effect of the LPS/galectin-3-Rag GTPases/Ragulator-mTORC1 axis on diabetogenesis is balanced with its bad effect on carcinogenesis? Moreover, the current study is limited by the availability of the diabetes and cancer data with respect to the LPS/galectin-3-Rag GTPases/Ragulator-mTORC1 axis. Nevertheless, our study supports a notion that galectin-3 is a sensor of LPS, an important modulator of the mTORC1 signaling, and a critical regulator of glucose metabolism, adding to the understanding of the molecular insights into diabetogenesis and tumorigenesis.

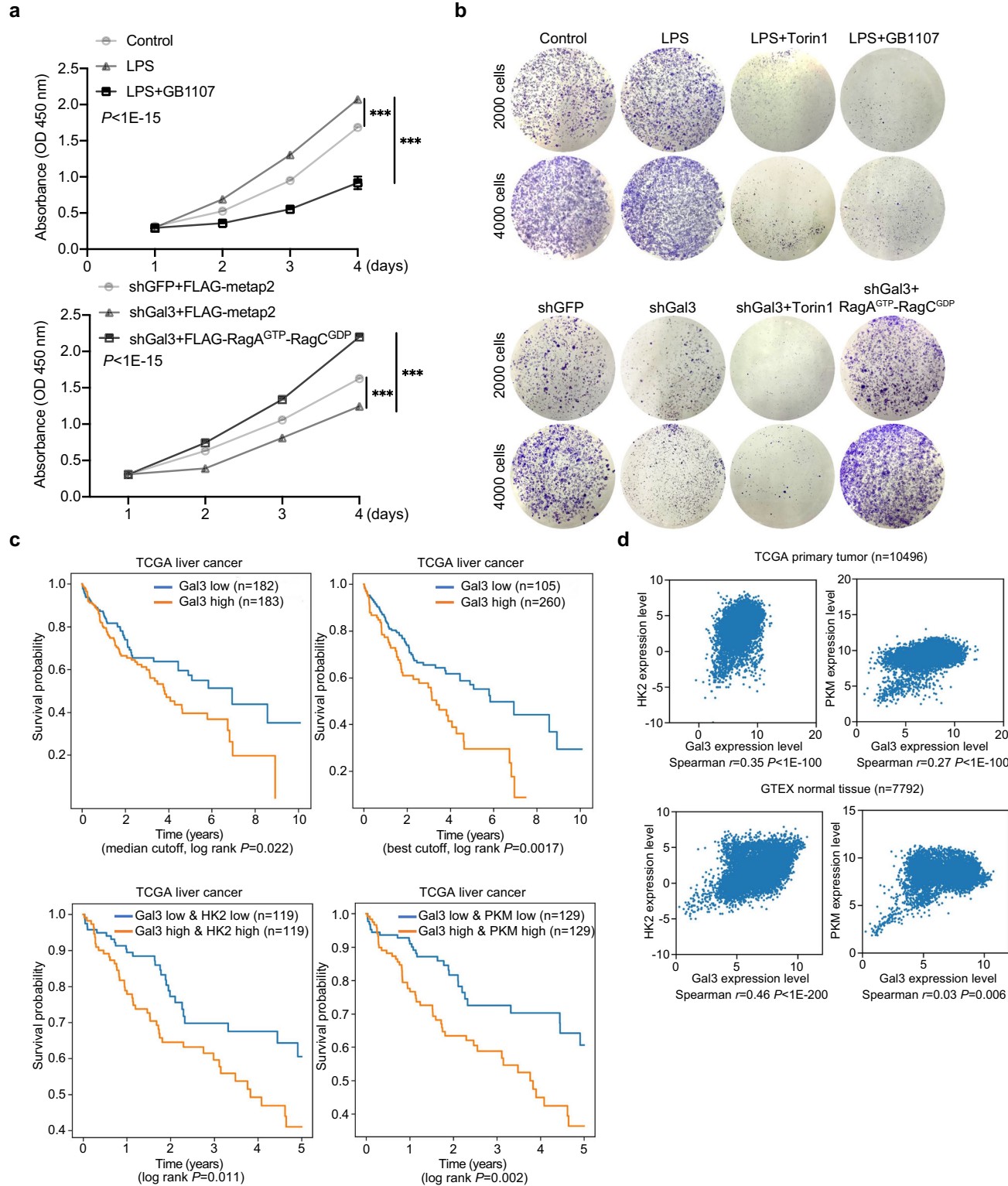

**Fig. 7 | Galectin-3 is implicated in hepatocarcinogenesis. a** CCK-8 assays for the proliferation of HepG2 cells were treated with LPS or LPS and GB1107, or infected with lentiviruses carrying shGFP or shGal3 and transfected with FLAG-metap2 or FLAG-RagA$^{GTP}$/RagC$^{GDP}$. Each bar represents the mean ± SD for biological triplicate experiments. *P* values were calculated by two-way ANOVA (\*\*\**P* < 0.001). **b** HepG2 cells were treated with LPS, LPS and Torin1, or LPS and GB1107, or infected with lentiviruses carrying shGFP or shGal3 and transfected with FLAG-RagA$^{GTP}$/RagC$^{GDP}$ in the presence or absence of Torin1 and cultured for 10 days before staining with crystal violet for colony formation assays. Representative images from biological triplicate experiments are shown. **c** Kaplan–Meier analysis for the correlation between survival time and galectin-3 signatures in HCC. *P* values were calculated by log-rank test. **d** Transcriptomes of TCGA primary tumor samples and GTEX normal tissue samples obtained from UCSD Xena for correlation analysis of the expression between galectin-3 and HK2 or PKM. Spearman correlation coefficients and *P* values were calculated by two-sided permutation test. **e** Kaplan-Meier analysis for the correlation between survival time and the expression levels of galectin-3 and HK2 or galectin-3 and PKM in HCC. *P* values were calculated by log-rank test.

## Methods

### Antibodies and reagents

Antibodies used: αRagA (4357, 1:1000 for Western Blotting (WB)), αRagC (3360,1:1000 for WB and 1:100 for Immunofluorescence (IF)), αmTOR (2972 and 2983, 1:1000 for WB and 1:100 for IF), αraptor (2280, 1:500 for WB), α4E-BP1 (9644, 1:1000 for WB), αp-4E-BP1 (2855, 1:1000 for WB), αTSC2 (4308, 1:1000 for WB), αS6K1 (9202, 1:1000 for WB), αp-S6K1 (9205, 1:500 for WB), αNPRL2 (37344, 1:1000 for WB), αWDR59 (53385, 1:1000 for WB), αp18 (8975, 1:1000 for WB and 1:100 for IF), αp14 (8145, 1:1000 for WB), αMyc (2272, 1:5000 for WB), αFLAG (14793, 1:10000 for WB), αHK2 (2867, 1:1000 for WB), αGPI (94068, 1:1000 for WB), αLDHA (3582, 1:1000 for WB), αULK1 (8054, 1:1000 for WB), αPKM1 (7067, 1:1000 for WB), αPKM2 (4053, 1:1000 for WB), and Normal Rabbit IgG (2729, 1:100 for Immunoprecipitation) from Cell Signaling; αmTOR (ab32028, 1:100 for IF), αLamp2 (ab25631, 1:100 for IF), αATP6V1B2 (ab73404, 1:1000 for WB), αSLC38A9 (ab81687, 1:1000 for WB), αGalectin-3 (ab209344, 1:1000 for WB), αGalectin-4 (ab175185, 1:1000 for WB), αGalectin-2 (ab232703, 1:1000 for WB), α*E. coli* LPS (ab35654, 1:100 for IF), αGLUT1 (ab115730, 1:1000 for WB), and αPFKM (ab204131, 1:1000 for WB) from Abcam; αWDR24 (20778-1-AP, 1:1000 for WB) and αSEC13 (15397-1-AP, 1:1000 for WB) from Proteintech; αβ-actin (AC004, 1:10000 for WB), HRP Goat Anti-Rabbit IgG (AS014, 1:5000 for WB), and HRP Goat Anti-Mouse IgG (AS003, 1:5000 for WB) from ABclonal; αHA (YM3146, 1:5000 for WB) from Biomed; αDEPDC5 (LS-C256035, 1:100 for WB) from LSBio. Control siRNA and siRNAs for Galectin-2, Galectin-3, Galectin-4, TSC2, and mouse galectin-3 as well as shRNAs for Galectin-3 were synthesized by Beijing Likely Biotechnology. *E. coli* LPS (L4391 and L3024), Torin1 (475991), Anti-FLAG M2 (A2220) affinity gel, and 3 × FLAG peptide (F3290) were from Sigma-Aldrich. Protein A or G beads (10004D) was from Invitrogen. Biotinylated LPS (tlrl-lpsbiot) was from InvivoGen. GB1107 (C108365) was from ChemeGen. Protease inhibitor cocktail was from Roche Applied Science. RPMI 1640 Medium Modified w/o Amino acids and Glucose (R9010-01) was from USBiological.

### Cell culture

The cell lines used were obtained from the American Type Culture Collection (ATCC). HEK 293 T cells were maintained in Dulbecco's modified Eagle's medium (DMEM) supplemented with 10% fetal bovine serum (FBS), HepG2 cells were maintained in DMEM and 15% FBS, and MEFs cells were maintained in DMEM with 20% FBS. All these cells were cultured in a humidified incubator equilibrated with 5% CO2 at 37 °C.

### Transfection

For transfection of cDNA expression constructs, 2-2.5 million cells were seeded in 15-cm dishes and transfected at 24 h after seeding. Experiments were done 36–48 h after transfection. For LPS treatment, LPS or DOTAP (Sigma) was suspended in OptiMEM allowing to equilibrate at room temperature for 5 min. Suspensions were then mixed and incubated for 30 min at room temperature and added slowly to the cultured cells. siRNA Transfections were carried out using Lipofectamine® RNAiMAX Reagent (Invitrogen, Carlsbad, CA) according to the manufacturer's instructions. The final concentration of siRNA used for each six-well plate was 50 nM. Each experiment was performed in triplicate and repeated at least three times. For RNAi experiment, at least three independent siRNA sequences were tested for each gene, and the one with the best efficiency was chosen. The siRNA sequences are as follows: Control siRNA: UUCUCCGAACGUGUCACGU; Galectin-3 siRNA-1: GCUUCAAUGAGAACAACAUT; Galectin-3 siRNA-2: CGGUGA AGCCCAAUGCAAAUT; Galectin-3 siRNA-3: CAUCAAUAUCCCUCUU GUATT; Galectin-2 siRNA-1: CCAUUGUCUGCAACUCAUUTT; Galectin-2 siRNA-2: UGUGUAAUAUUCUAGGAUAUT; Galectin-2 siRNA-3: UGU CCUCUUUCAAGUUAAATT; Galectin-4 siRNA-1: GAGGGUGGACACA UUGGAATT; Galectin-4 siRNA-2: GCUUCAAGGUUUACGCCAAUT;

Galectin-4 siRNA-3: GCAAGAGCUUUGCUAUCAAUT; galectin-3 siRNA-1: GGAGAGUCAUUGUGUGUAAUT; galectin-3 siRNA-2: ACACGAAGCA GGACAAUAAUT; galectin-3 siRNA-3: GAGAACAACAGGAGAGUCAUT; raptor siRNA-1: ACAAUGUGGGCUUAUUGGGUC; raptor siRNA-2: AUCCUUAUCUCUAACUCUGAG; raptor siRNA-3: AAUUUGCACCGAU GGUUUCCA; ULK1 siRNA-1: UGUUUUCAUGUUUCAGUUCCU; ULK1 siRNA-2: UGUUCUUCUCGUAGAACAGGC; ULK1 siRNA-3: UCUCUA UAUGCAUAAAGUGCU; TSC2 siRNA-1: CGAACGAGGUGGUGUCCUA; TSC2 siRNA-2: GGAUUACCCUUCCAACGAA; TSC2 siRNA-3: GCACGA UGACAUCAUGCAA.

### Immunoprecipitation and western blotting

Cellular extracts were prepared by incubating about $5 \times 10^8$ cells in lysis buffer (50 mM Tris-HCl, pH 8.0, 150 mM NaCl, 0.5% NP40, and 1 tablet of EDTA-free protease inhibitor [Roche] (per 25 ml buffer)) for 30 min at 4 °C. This was followed by centrifugation at $16,000 \times g$ for 15 min at 4 °C. For immunoprecipitation, 500 μg of protein was incubated with specific antibodies (2-3 μg) for 12 h at 4 °C with constant rotation; 60 μl of 50% protein G agarose beads was then added and the incubation was continued for an additional 2 h. Beads were then washed 5 times using the lysis buffer. The precipitated proteins were eluted from the beads by resuspending the beads in 2 × SDS-PAGE loading buffer and boiling for 10 min. The resultant materials from immunoprecipitation or cell lysates were resolved using 8%-12% SDS-PAGE gels and transferred onto acetate cellulose membranes. For anti-FLAG immunoprecipitation, anti-FLAG M2 Affinity Gel (Sigma) was washed with lysis buffer three times then resuspended to a ratio of 50:50 affinity gel to lysis buffer before 25 μl of a well-mixed slurry was added to cleared lysates and incubated at 4 °C in a shaker for 90–120 min. Immunoprecipitated proteins were denatured by the addition of 2 × SDS-PAGE loading buffer and boiled for 10 min. Denatured samples were resolved by 8–12% SDS-PAGE, and analyzed by immunoblotting. For western blotting analysis, membranes were incubated with appropriate antibodies for overnight at 4 °C followed by incubation with a secondary antibody. Immunoreactive bands were visualized using western blotting Luminal reagent (Santa Cruz Biotechnology) according to the manufacturer's recommendation.

### Silver staining and mass spectrometry

HEK-293T cells e*x*pressing FLAG-Gal3, FLAG-RagA, or FLAG-RagC were washed twice with cold PBS, scraped, and collected by centrifugation at $800 \times g$ for 5 min. Cellular extracts were prepared by incubating the cells in lysis buffer containing protease inhibitor cocktail (Roche). Anti-FLAG immunoaffinity columns were prepared using anti-FLAG M2 affinity gel (Sigma) following the manufacturer's suggestions. Cell lysates were obtained from about $5 \times 10^8$ cells and applied to an equilibrated FLAG column of 1-mL bed volume to allow for adsorption of the protein complex to the column resin. After binding, the column was washed with cold PBS plus 0.1% Nonidet P-40 prior to application of 3x FLAG peptides to elute FLAG protein complex as described by the vendor. Fractions of the bed volume were collected and resolved on NuPAGE 4–12% Bis-Tris gel (Invitrogen), silver-stained using Pierce Silver Stain Kit, and subjected to LC/MS-MS sequencing. Peptides recovered by digestion of proteins within the gel was loaded onto an Acclaim PePmap C18-reversed phase column (75 μm × 2 cm, 3 μm, 100 Å, Thermo scientific) and separated with reversed phase C18 column (75 μm × 10 cm, 5 μm, 300 Å, Agela Technologies) mounted on a Dionex ultimate 3000 nano LC system. Peptides were eluted using a gradient. The eluates were directly entered Q-Exactive MS (Thermo Fisher Scientific, Waltham, MA, USA), setting in positive ion mode and data-dependent manner with full MS scan from 350–2000 *m/z*, full scan resolution at 70,000, MS/MS scan resolution at 17,500. MS/MS scan with minimum signal threshold 1E + 5, isolation width at 2 Da. Peptide identification and quantification was carried out on the Mascot software

Revision 2.3.01 using the TAIR database search algorithm and the integrated false discovery rate (FDR) analysis function.

## FPLC chromatography

Cells proteins were extracted by anti-FLAG M2 Affinity Gel. Approximately 6 mg protein was concentrated to 1 ml using a Millipore Ultrafree centrifugal filter apparatus (3 kDa molecular mass limit), and then applied to an 850 × 20 mm Superose 6 size exclusion column (Amersham Biosciences) that had been equilibrated with dithiothreitol-containing buffer and calibrated with protein standards (blue dextran, 2000 kDa; thyroglobulin, 669 kDa; ferritin, 440 kDa; and aldolase, 158 kDa; all from Amersham Biosciences). The column was eluted at a flow rate of 0.5 ml/min and fractions were collected.

## Pull-down assays

GST-fused constructs were expressed in BL21 *Escherichia coli*. In vitro transcription and translation experiments were done with rabbit reticulocyte lysate (TNT systems, Promega) according to the manufacturer's recommendation. In GST pull-down assays, about 5 μg of the appropriate GST fusion proteins with 30 μl of glutathione-Sepharose beads was incubated with 5–8 μl of in vitro transcribed/translated products in binding buffer (75 mM NaCl, 50 mM HEPES, pH 7.9) at 4 °C for 2 h in the presence of the protease inhibitor mixture. The beads were washed 5 times with binding buffer, resuspended in 30 μl of 2 × SDS-PAGE loading buffer, and detected by western blotting. For Pull-Down assay of biotinylated LPS, the reaction mixtures containing recombinant galectin-3 and biotinylated LPS were incubated for 4 h at 4 °C. Pull-down of LPS was carried out by treating the mixtures with streptavidin beads (Cytiva).

## Fluorescence confocal microscopy

HEK-293T cells growing on 6-well chamber slides were washed with PBS, fixed in 4% paraformaldehyde, permeabilized with 0.2% Triton X-100, blocked with 0.8% BSA, and incubated with appropriate primary antibodies followed by addition of Alexa Fluor™ 488/568 donkey secondary antibodies (Invitrogen). DAPI (Sigma) was included in the final wash to stain nuclei. Images were visualized with an Olympus inverted microscope equipped with a charge coupled camera.

## RNA-seq

Total RNAs were extracted from HEK-293T cells and subjected to deep sequencing by BGI (LC-Bio, Hangzhou) through BGI500 sequencer with single-end 50-bp reads. Raw data were preprocessed through fastp to remove adaptors and low-quality reads with the default parameters. Clean reads were aligned to GRCh38/hg38 reference genome through STAR, an ultrafast universal RNA-seq aligner. Raw read count mapped to every gene that obtained via HTSeq-Count tool was used as expression level. DESeq2 R Bioconductor package was used for screening differential expression genes with the threshold of $P < 0.05$ and absolute log 2 (fold change) > 1.

## Glucose consumption and lactate production assay

Glucose Uptake Colorimetric Assay Kits and Lactate Colorimetric Assay Kits (Sigma-Aldrich Corporation) were used, and glycolysis was detected in HepG2 or HEK-293T cells according to the manufacturer's protocols.

## Cell viability/proliferation assay

For cell proliferation assays, HepG2 cells were seeded into 96-well plates with an equal volume of medium. LPS (1 μg/ml) or LPS (1 μg/ml) plus GB1107 (0.5 μm) was added, or plasmids expressing different proteins were transfected. After cells treatment, 10 μl CCK-8 solution was added according to the manufacturer's protocol. Plates were incubated at 37 °C for 2 h and cell viability was determined by measuring the absorbance at 450 nm wavelength. Each experiment was performed in triplicate and repeated at least three times.

## qPCR

Total cellular RNAs were isolated with the TRIzol reagent (Invitrogen) and used for the first strand cDNA synthesis with the Reverse Transcription System (Roche). Quantitation of all gene transcripts was done by real time RT-PCR (qPCR) using Power SYBR Green PCR Master Mix and Roche LightCycler®480 II sequence detection system. The qPCR primers sequences were: HK2: GAGCCACCACTCACCCTACT (F), CCAGGCATTCGGCAATGTG (R); PKM2: ATGTCGAAGCCCCATAGTGAA (F), TGGGTGGTGAATCAATGTCCA (R); GLUT1: GGCCAAGAGTGTGCTAAAGAA (F), ACAGCGTTGATGCCAGACAG (R); GPI: CAAGGACCGCTTCAACCACTT (F), CCAGGATGGGTGTGTTTGACC (R); LDHA: ATGGCAACTCTAAAGGATCAGC (F), CCAACCCCAACAACTGTAATCT (R); PFKM: GGTGCCCGTGTCTTCTTTGT (F), AAGCATCATCGAAACGCTCTC (R); Galectin-3: ATGGCAGACAATTTTTCGCTCC (F), GCCTGTCCAGGATAAGCCC (R).

## Lentiviral production and infection

Recombinant lentiviruses expressing shGFP, shGal3, GFP-Gal3, HA-metap2, and HA-Gal3 were constructed by Beijing Likely Biotechnology. Concentrated viruses were used to infect $5 × 10^5$ cells in a 60-mm dish with 8 μg/ml polybrene. Infected HEK-293T or HepG2 cells were then subjected to sorting target expression. The shRNA sequences were: shGFP: UUCUCCGAACGUGUCACGU; shGal3-1: GCCACTGATTGTGCCTTATAA; shGal3-2: GCATGCTGATAACAATTCTGG; shGal3-3: GGAGAGTCATTGTTTGCAATA.

## Colony formation assay

HepG2 cells were maintained in culture media in 6-well plate for 10 days, fixed with 4% paraformaldehyde, and then stained with crystal violet. Each experiment was performed in triplicate and repeated at least three times.

## Bioinformatics

The RNA-seq expression data and overall survival data of TCGA liver cancer are downloaded from UCSC Xena; the median expression levels of galectin-3 and HK2 were used as the cutoff to classify high and low expression samples. The median/best cutoff of galectin-3 in TCGA liver cancer and related survival data were collected from Human Protein Atlas (https://www.proteinatlas.org/ENSG00000131981-LGALS3/pathology/liver+cancer).

## Statistical analysis

Data from biological triplicate experiments are presented. An unpaired, two-tailed Student's $t$ test was used for 2-group comparisons. ANOVA with Bonferroni's correction was used to compare multiple groups. A $P$ value of less than 0.05 was considered statistically significant. Statistical results were determined using Prism 8 software or Scipy package. Before statistical analysis, variations within each group and the assumptions of the tests were checked.

## Reporting summary

Further information on research design is available in the Nature Portfolio Reporting Summary linked to this article.

# Data availability

The RNA-seq data generated in this study have been deposited in the NCBI Gene Expression Omnibus (GEO) under accession code GSE211784. Mass spectrometry data are provided in Supplementary Data 1. Computational analysis of diabetes patient data from ArrayExpress (E-MEXP-2559). Datasets of "thyroid gland papillary carcinoma", "prostatic intraepithelial neoplasia", and "hepatocellular carcinoma" from Oncomine database (https://www.oncomine.org).

The median/best cutoff of galectin-3 in TCGA liver cancer and related survival data were collected from Human Protein Atlas (https://www.proteinatlas.org/ENSG00000131981-LGALS3/pathology/liver+cancer). All data generated or analyzed during this study are included in this published article (and its supplementary information files). The reporting summary and editorial checklist for this article are available as a Supplementary File. Source data are provided with this paper.

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

## Acknowledgements

This work was supported by a grant (2021YFA1300603 to Y.S.) from the Ministry of Science and Technology of China, and grants (82188102, 81730079, and 31991164 to Y.S.) from the National Natural Science Foundation of China.

## Author contributions

X.C. and Y.S. conceived the project and designed the experiments; X.C., B.L., X.W., Jiajing Wu, and Y.W. performed experiments; X.C., C.Y., and X.L. analyzed data; D.Y., L.H., Z.T., X.Y., Jianqiu Wang, S.L., and L.S. provided technical assistance; X.C. and Y.S. wrote the manuscript.

## Competing interests

The authors declare no competing interests.
