## [Peer Review File · Nature Communications]

Intracellular Galectin-3 Is a Lipopolysaccharide Sensor that Promotes Glycolysis through mTORC1 ActivationREVIEWER COMMENTS

Reviewer #1 (Remarks to the Author):

Chen et al. reported a novel role of galectin-3 in regulating MTORC1. They show galectin-3 can sense LPS to activate MTORC1 signaling for regulating glucose metabolism. Even though, how galectin-3 transduces LPS signaling and how LPS affects MTORC1 is not very clear, the collection of interesting observations and mechanistic studies in this manuscript brought an interesting story and laid the foundation for further explorations. Below are a few questions that may help to clarify some details further.

1. Figure 1E, the quality of immunofluorescent image is low. GFP-Gal3 seems to behave differently in each case; LAMP2/Ragc puncta is hard to tell; the zoom-in region should be pointed out. Also, Gal3 is cytosolic protein under normal condition, and forms puncta on lysosomes when lysosomal membrane is damaged, which has been reported in many studies (e.g. PMID: 22246324; 26114578; 31813797). How do authors explain it? Is there endomembrane damage in this case? It needs to be clarified.
2. Figure 1F, the lysosomal proteins should also be tested, which can strengthen the result here.
3. Figure 1H/2G, Gal8 has been showed to associate with MTORC1 complex during lysosomal damage (PMID: 29625033); the most universal expressed galectins are Gal1 and Gal3 (PMID: 26114578). What is the reason to test Gal2 and Gal4 here?
4. Figure 2A, left panel of IP: the FLAG-RagC of second lane is weaker than it in third lane; right panel: the increase is not obvious. So, quantification of these experiments should be included.
5. Figure 2B, is metap2/RagA/C tagged with FLAG? It seems to be mislabeled.
6. Figure2, what is the physiological meaning of interaction between Rag GTPases and Ragulator in the mTORC1 signaling? the status of Rag nucleotide loading by Ragulator as guanine nucleotide exchange factor is needed to be explained here.
7. Figure 3, does LPS affect MTORC1 via Rag/Ragulator system? Or other MTORC1 effectors, such as TSC2 pathway?
8. Is the carbohydrate binding ability of Gal3 required to sense LPS? Testing the R186 site mutant of Gal3 might answer this question.
9. Figure 5/6, is that possible that the decrease level of HK2, PKM2, and GLUT1 in Gal3 knockdown cells is caused by the activation of degradation pathways, such as autophagy. Since MTOR activity is downregulated in Gal3 knockdown cells, which can lead to initiation of autophagy for degradation.
10. Can overexpressed Gal3 further boost MTORC1 activity during LPS treatment?

Reviewer #2 (Remarks to the Author):

Chen et al investigated the role of intracellular galectin-3 and found that it is required for mTORC1 activation in response to LPS. First, they showed that overexpressed Gal3, as well as endogenous Gal3, was associated with RagA and other components of mTORC1 signaling and the lysosomal surface. Second, upon knockdown of Gal3 expression, they found decreased mTORC1 signaling and association of Rag with p14. Third, they showed that the LPS-induced mTORC1 activation was inhibited by Gal3 depletion or pharmacological inhibition. Increased expression of Gal3 also further augmented the LPS-induced mTORC1 activation.

Furthermore, by Gal3 knockdown vs mTOR inhibition, they provided correlation that Gal3 (like mTOR) is involved in promoting glycolysis. Lastly, by mining public datasets, they found that Gal3 is overexpressed in HCC and that depletion of Gal3 prevented colony formation of HepG2 cells and that this can be rescued by overexpression of RagA/C. Based on these findings they concluded that "Intracellular Galectin3 is a sensor of lipopolysaccharide that promotes glycolysis through activating mTORC1".

The studies are interesting and the data are quite convincing. The finding that Gal3 is involved in the LPS-induced mTORC1 activation also provides new insights/mechanisms as to how mTORC1 is activated by this stimulus. The paper is also presented well.

There are some issues that would need to be addressed as follows:

Main issues:

1. The authors should examine how deletion of the Gal3 carbohydrate recognition domain affect LPS-induced mTORC1 signaling.

2. The authors propose that Gal3 "senses" LPS. So far, the data suggest that Gal3 is required for the LPS-induced S6K1 phosphorylation. A sensor would imply that Gal3 will react to levels or amounts of the nutrient/metabolite to modulate mTORC1 signaling. The authors should conduct titration studies to determine how increasing LPS (with same Gal3 expression) or Gal3 levels (using same LPS concentration) affect mTORC1 activation.

3. Torin1 also inhibits mTORC2. In order to link the role of Gal3 in promoting glycolysis via mTORC1, they should knockdown raptor (eg in Fig 5G).

Other comments:

Fig. 2F: There is a big bubble on pS6K1 blot

Fig 2D: Gal3 depletion somewhat diminishes expression of raptor and S6K (see without LPS addition). The effect of gal3 depletion upon LPS addition on pS6K1 is not so robust given less expression of S6K1 upon gal3 depletion. They should quantitate this with standard deviation.

Fig. 3C: They should also analyze 4EBP phosphorylation upon Gal3 knockdown and overexpression.

Fig. 4: Will knockdown of Gal3 diminish the LPS/Rag association?

Does the absence of Gal3 decrease LPS-induced Rag GTP loading?

Some labeling of blots are not properly aligned (eg Fig 4F, 4G)

Reviewer #3 (Remarks to the Author):

Galectin-3 is a member of the β -galactoside-binding protein family and is reported to be involved in cellular metabolism and linked to diabetes and cancer. In this manuscript, the Shang team performed IP-MS to identify proteins that bind to galectin-3. Some of these interacting proteins (Rag GTPases and Regulator of the mTORC1 signaling machinery) were validated using Co-immunoprecipitation, FPLC and GST pull-down. The functional study showed that galectin-3 senses LPS to facilitate the interaction of Rag GTPases and Regulator. They demonstrated that galectin-3 is not only as a sensor of LPS, but also as a regulator of glucose metabolism and tumorigenesis. The text is well written and the authors present a valuable compilation of information about the functions of galectin-3 and give new insights into the molecular mechanism of galectin-3 in diabetogenesis and tumorigenesis. I recommend that the manuscript be accepted pending the following revisions:

1. Regarding immunopurification and mass spectrometry analysis, the number of samples used in IP and MS experiments is not clearly described. How many replicate samples were collected for MS identification? The MS analysis is only minimally described. The authors should give more details of LC gradient and MS parameters. All MS raw data and annotated, mass labeled spectra for all identified peptides must be provided in a publicly accessible database (and database reference/entry number is

provided) or submitted as supplemental material with the manuscript. Without having access to the raw data or peak lists I cannot tell the quality of the MS experiments.

2. Figure 2. The authors used MEFs cells (mouse embryonic fibroblasts) for the experiment (Figure 2D). Please explain the reason why MEFs cells were used in this manuscript.

3. Figure 2C-D. Inconsistent with results in HEK-293T cell lines, knockdown of galectin-3 in MEFs cells reduced both the protein and phosphorylation levels of S6K1. The authors need to clarify this point.

3. Figure 7A. The statistical significance of different groups should be analyzed and indicated in the figure.

4. Figure 7B. How many the biological repeats were carried out? The statistical analysis is required.

5. Line 648, Line 678. These sentences should be checked and amended.

6. Please include the detailed description of TCGA data analysis.

7. What was the concentration of siRNA used or transfected in this manuscript? This should be written clearly in the experimental procedures section.

8. Some Western blotting results lack internal controls bands (GAPDH or β -actin etc). The authors are suggested to provide the results of control experiments.

Re: Nature Communications MS# NCOMMS-22-21714-T

Title: “Intracellular Galectin-3 Is a Sensor of Lipopolysaccharide that Promotes Glycolysis through Activating mTORC1”

Response to Reviewer #1’s comments:

Comments:

Chen et al. reported a novel role of galectin-3 in regulating MTORC1. They show galectin-3 can sense LPS to activate MTORC1 signaling for regulating glucose metabolism. Even though, how galectin-3 transduces LPS signaling and how LPS affects mTORC1 is not very clear, the collection of interesting observations and mechanistic studies in this manuscript brought an interesting story and laid the foundation for further explorations. Below are a few questions that may help to clarify some details further.

1. Figure 1E, the quality of immunofluorescent image is low. GFP-Gal3 seems to behave differently in each case; LAMP2/Ragc puncta is hard to tell; the zoom-in region should be pointed out. Also, Gal3 is cytosolic protein under normal condition, and forms puncta on lysosomes when lysosomal membrane is damaged, which has been reported in many studies (e.g. PMID: 22246324; 26114578; 31813797). How do authors explain it? Is there endomembrane damage in this case? It needs to be clarified.

Authors: To address the reviewer's concerns, we have repeated the immunofluorescence experiments on subcellular co-localization of Gal3 with Lamp2, RagC, and p18 and replaced the image (Fig. 1e). As the reviewer rightfully points out, Gal3 forms puncta in lysosomes after lysosomal damage, as do other members of the Galectins family (PMID: 29625033; 30081722). As our experiments were performed under standard cell culture conditions, we did not expect endomembrane damages. However, we appreciate the reviewer’s point and have added it to discussion.

2. Figure 1F, the lysosomal proteins should also be tested, which can strengthen the result here.

Authors: The detection of the lysosomal protein Lamp2 has been added to the Fig. 1f.

3. Figure 1H/2G, Gal8 has been showed to associate with mTORC1 complex during lysosomal damage (PMID: 29625033); the most universal expressed galectins are Gal1 and Gal3 (PMID: 26114578). What is the reason to test Gal2 and Gal4 here?

Authors: Galectins are classified as prototypical, chimera, and Tandem Repeat. Since Gal3 represents chimera, we thus tested Gal2 as a prototypical and Gal4 as a Tandem Repeat for comparison. The point has been clarified in the revision.

4. Figure 2A, left panel of IP: the FLAG-RagC of second lane is weaker than it in third lane; right panel: the increase is not obvious. So, quantification of these experiments should be included.

Authors: To comply with the reviewer, we have quantified the results in the revision (Supplementary Fig. 1a).

5. Figure 2B, is metap2/RagA/C tagged with FLAG? It seems to be mislabeled.

Authors: The reviewer is correct. We appreciate it and have corrected the label.

6. Figure 2, what is the physiological meaning of interaction between Rag GTPases and Ragulator in the mTORC1 signaling? the status of Rag nucleotide loading by Ragulator as guanine nucleotide exchange factor is needed to be explained here.

Authors: Rag GTPases recruit mTORC1 to the lysosomal surface and activate mTORC1 in the presence of amino acids (PMID:18497260; 23263183), where Ragulator is required as a lysosomal scaffold for Rag GTPases and as a guanine nucleotide exchange factor (GEF) (PMID:22980980; 20381137) for activating RagA or RagB (RagA/B). The interaction between Rag GTPases and Ragulator, we propose, is facilitated by Galectin-3. This has been discussed further in the revision.

7. Figure 3, does LPS affect mTORC1 via Rag/Ragulator system? Or other mTORC1 effectors, such as TSC2 pathway?

Authors: Loss-of-function of TSC2 experiments in Fig. 4i show that LPS affects mTORC1 via Rag/Ragulator system, but not the TSC2 pathway. This has been clarified in the revision.

8. Is the carbohydrate binding ability of Gal3 required to sense LPS? Testing the R186 site mutant of Gal3 might answer this question.

Authors: Both the N-terminal domain (1-107aa) and the C-terminal carbohydrate recognition domain (108-250aa or Gal3^{R186S}) of Gal3 are required for the activation of the mTORC1 signaling pathway by LPS. The results are shown in Supplementary Fig. 2b in the revision.

9. Figure 5/6, is that possible that the decrease level of HK2, PKM2, and GLUT1 in Gal3 knockdown cells is caused by the activation of degradation pathways, such as autophagy. Since mTOR activity is downregulated in Gal3 knockdown cells, which can lead to initiation of autophagy for degradation.

Authors: We appreciate the reviewer for this point. Accordingly, galectin-3 and ULK1 were co-knocked down in HEK-293T cells, and we found that autophagy inhibition could not restore the expression of HK2, PKM2 and GLUT1 (Supplementary Fig. 3a), arguing against the possibility that the decrease level of HK2, PKM2, and GLUT1 in Gal3-deficient cells is caused by the activation of autophagy for degradation.

10. Can overexpressed Gal3 further boost mTORC1 activity during LPS treatment?

Authors: Overexpression of Gal3 further enhanced mTORC1 activity during LPS treatment, as demonstrated by the experimental results in Fig. 3c and Fig. 6f.

Re: Nature Communications MS# NCOMMS-22-21714-T

Title: “Intracellular Galectin-3 Is a Sensor of Lipopolysaccharide that Promotes Glycolysis through Activating mTORC1”

Response to Reviewer #2’s comments:

Comments:

Chen et al investigated the role of intracellular galectin-3 and found that it is required for mTORC1 activation in response to LPS. First, they showed that overexpressed Gal3, as well as endogenous Gal3, was associated with RagA and other components of mTORC1 signaling and the lysosomal surface. Second, upon knockdown of Gal3 expression, they found decreased mTORC1 signaling and association of Rag with p14. Third, they showed that the LPS-induced mTORC1 activation was inhibited by Gal3 depletion or pharmacological inhibition. Increased expression of Gal3 also further augmented the LPS-induced mTORC1 activation. Furthermore, by Gal3 knockdown vs mTOR inhibition, they provided correlation that Gal3 (like mTOR) is involved in promoting glycolysis. Lastly, by mining public datasets, they found that Gal3 is overexpressed in HCC and that depletion of Gal3 prevented colony formation of HepG2 cells and that this can be rescued by overexpression of RagA/C. Based on these findings they concluded that “Intracellular Galectin3 is a sensor of lipopolysaccharide that promotes glycolysis through activating mTORC1”.

The studies are interesting and the data are quite convincing. The finding that Gal3 is involved in the LPS-induced mTORC1 activation also provides new insights/mechanisms as to how mTORC1 is activated by this stimulus. The paper is also presented well.

There are some issues that would need to be addressed as follows:

Main issues:

1. The authors should examine how deletion of the Gal3 carbohydrate recognition domain affect LPS-induced mTORC1 signaling.

Authors: To comply with the reviewer request, experiments were performed with Gal-3 mutants via deletion or point mutation. We found that both the N-terminal domain (1-107aa) and the C-terminal carbohydrate recognition domain (108-250aa or Gal3^{R186S}) of Gal-3 are required for the activation of the mTORC1 signaling pathway by LPS. The results are shown in Supplementary Fig. 2b in the revision.

2. The authors propose that Gal3 “senses” LPS. So far, the data suggest that Gal3 is required for the LPS-induced S6K1 phosphorylation. A sensor would imply that Gal3 will react to

levels or amounts of the nutrient/metabolite to modulate mTORC1 signaling. The authors should conduct titration studies to determine how increasing LPS (with same Gal3 expression) or Gal3 levels (using same LPS concentration) affect mTORC1 activation.

Authors: We appreciate the reviewer for this point and performed titration studies to further explore the effect of LPS on mTORC1 signaling pathway. The results showed that the phosphorylation of S6K1 increased with increasing concentration of LPS, when Gal3 expression was about the same (Supplementary Fig. 2a).

3. Torin1 also inhibits mTORC2. In order to link the role of Gal3 in promoting glycolysis via mTORC1, they should knockdown raptor (eg in Fig 5G).

Authors: We thank the reviewer for pointing this out. Experiments were performed by knocking down raptor, and the results support our argument (Supplementary Fig. 3b).

Other comments:

Fig. 2F: There is a big bubble on pS6K1 blot.

Authors: We have repeated the experiments and replaced the blot in Fig. 2f.

Fig 2D: Gal3 depletion somewhat diminishes expression of raptor and S6K (see without LPS addition). The effect of gal3 depletion upon LPS addition on pS6K1 is not so robust given less expression of S6K1 upon gal3 depletion. They should quantitate this with standard deviation.

Authors: The data have been quantitated and statistical analysis has been provided in Supplementary Fig. 1b.

Fig. 3C: They should also analyze 4EBP phosphorylation upon Gal3 knockdown and overexpression.

Authors: The phosphorylation level of 4E-BP1 has been analyzed upon knockdown or overexpression of Gal3.

Fig. 4: Will knockdown of Gal3 diminish the LPS/Rag association?

Authors: Based on the results of our experiments, we propose that LPS influences mTORC1 signaling pathway by binding Gal3; it does not interact with Rag GTPases.

Does the absence of Gal3 decrease LPS-induced Rag GTP loading?

Authors: As stated above, our results support a theme where LPS binds to Gal3, facilitating the interaction between Rag GTPases and Ragulator. Thus, the absence of Gal3 affects the interaction, but not the loading of Rag GTPases.

Some labeling of blots are not properly aligned (eg Fig 4F, 4G)

Authors: We appreciate the reviewer for this point and have adjusted the label of the blots.

Re: Nature Communications MS# NCOMMS-22-21714-T

Title: “Intracellular Galectin-3 Is a Sensor of Lipopolysaccharide that Promotes Glycolysis through Activating mTORC1”

Response to Reviewer #3’s comments:

Comments:

Galectin-3 is a member of the β -galactoside-binding protein family and is reported to be involved in cellular metabolism and linked to diabetes and cancer. In this manuscript, the Shang team performed IP-MS to identify proteins that bind to galectin-3. Some of these interacting proteins (Rag GTPases and Ragulator of the mTORC1 signaling machinery) were validated using Co-immunoprecipitation, FPLC and GST pull-down. The functional study showed that galectin-3 senses LPS to facilitate the interaction of Rag GTPases and Ragulator. They demonstrated that galectin-3 is not only as a sensor of LPS, but also as a regulator of glucose metabolism and tumorigenesis. The text is well written and the authors present a valuable compilation of information about the functions of galectin-3 and give new insights into the molecular mechanism of galectin-3 in diabetogenesis and tumorigenesis. I recommend that the manuscript be accepted pending the following revisions:

1. Regarding immunopurification and mass spectrometry analysis, the number of samples used in IP and MS experiments is not clearly described. How many replicate samples were collected for MS identification? The MS analysis is only minimally described. The authors should give more details of LC gradient and MS parameters. All MS raw data and annotated, mass labeled spectra for all identified peptides must be provided in a publicly accessible database (and database reference/entry number is provided) or submitted as supplemental material with the manuscript. Without having access to the raw data or peak lists I cannot tell the quality of the MS experiments.

Authors: To comply with the reviewer’s request, we have added to method section more details about IP and MS experiments, including the number of samples, LC gradient, and MS parameters, and provided the detailed results of mass spectrometry analysis in Supplementary Table 1.

2. Figure 2. The authors used MEFs cells (mouse embryonic fibroblasts) for the experiment (Figure 2D). Please explain the reason why MEFs cells were used in this manuscript.

Authors: The majority of our experiments were performed in HEK-293T cells, a cancer cell line derived from human embryonic kidney. MEFs cells were also used for two considerations: 1) the representativeness of our observations in other tissues or cell lines; and 2) the physiological relevance of our observations.

3. Figure 2C-D. Inconsistent with results in HEK-293T cell lines, knockdown of galectin-3 in MEFs cells reduced both the protein and phosphorylation levels of S6K1. The authors need to clarify this point.

Authors: To address the reviewer's comment, we have repeated the experiment and quantified the results (Fig. 2c, 2d and Supplementary Fig. 1b).

3. Figure 7A. The statistical significance of different groups should be analyzed and indicated in the figure.

Authors: We have added the results of the statistical analysis in Fig. 7a.

4. Figure 7B. How many the biological repeats were carried out? The statistical analysis is required.

Authors: The experiment had three biological repeats. The results of the statistical analysis have been added (Supplementary Fig. 4).

5. Line 648, Line 678. These sentences should be checked and amended.

Authors: We appreciate the reviewer for bringing this to our attention and have modified the text.

6. Please include the detailed description of TCGA data analysis.

Authors: The detailed description of TCGA data analysis has been added to the method section.

7. What was the concentration of siRNA used or transfected in this manuscript? This should be written clearly in the experimental procedures section.

Authors: The detailed information has been added to the methods section.

8. Some Western blotting results lack internal controls bands (GAPDH or β -actin etc). The authors are suggested to provide the results of control experiments.

Authors: Internal controls have been added to Fig. 2c, 2d, 2h, 2g, Fig. 3a, 3b, 3d, Fig. 5e, 5f, and Fig. 6g, 6h.

Reviewer #1 (Remarks to the Author):

The authors have successfully addressed all raised questions and the manuscript is now significantly improved.

Reviewer #2 (Remarks to the Author):

Lines 222-223: “We also tested a series of biomolecules including glucose, lactose...” The data for these were not shown. Supp. Fig 2b: line 241: “absence of either part no longer activated mTORC1.” They should modify this sentence since the mutants only diminished the LPS-induced activation.

Supp. Fig 2a: The results of this experiment only suggest a dose-response activation of mTORC1 by LPS given endogenous equal levels of Gal3. Whether Gal3 levels affect mTORC1 activation under the same dose of LPS was not addressed. The claim that Gal3 senses LPS is important to carefully address since it is not clear whether Gal3 is binding to LPS intracellularly. So far, only in vitro pull-down data support this (but this would not be surprising since Gal3 is also present extracellularly). The interaction of Gal3 with the Rags under LPS-induced conditions was shown convincingly. However, whether this occurs via intracellular binding of LPS to Gal3 and thus whether Gal3 is “sensing” LPS remains obscure. The authors should either perform experiments to further address this issue (eg use of Gal3 mutants), or modify their conclusion as to the “sensor” nature of Gal3. The data are supportive of LPS-induced mTORC1 activation via Gal3 binding to Rags/Ragulator but do not provide convincing data on intracellular LPS binding to Gal3, hence the LPS sensor status of Gal3 is tenuous.

Fig. 4f:

Line 275-276: “we next investigated whether the interaction of galectin-3 with RagA/RagC and Ragulator influences the nucleotide state of Rag GTPases.”

Line 289-290: “...suggesting that the interaction of galectin-3 with RagA/RagC is functionally linked to the nucleotide state of RagA and the generation of active Rag GTPases.”

The experiments in Fig 4f examines how Gal3 binds to the Rags, ie their conclusion is that Gal3 prefers to bind to active Rags by virtue of their nucleotide state. Thus, Lines 275-276 should be modified. They did not address whether the Gal3 binding influences the nucleotide state of Rags. This is the reason why this reviewer inquired whether Gal3 absence could decrease Rag-GTP loading. The authors did not address this inquiry, however.

Reviewer #3 (Remarks to the Author):

I am satisfied with authors' new experimental data and their responses to my comments on the previous version.

REVIEWER #2:

Lines 222-223: “We also tested a series of biomolecules including glucose, lactose...” The data for these were not shown.

Authors: As we mentioned in our manuscript, the results for these biomolecules are negative and have no influence on the conclusion of the paper. So, we did not show these results.

Supp. Fig 2b: line 241: “absence of either part no longer activated mTORC1.” They should modify this sentence since the mutants only diminished the LPS-induced activation.

Authors: We appreciate the reviewer’s comment and have modified the sentence as “absence of either part diminished the LPS-induced mTOR activation”.

Supp. Fig 2a: The results of this experiment only suggest a dose-response activation of mTORC1 by LPS given endogenous equal levels of Gal3. Whether Gal3 levels affect mTORC1 activation under the same dose of LPS was not addressed. The claim that Gal3 senses LPS is important to carefully address since it is not clear whether Gal3 is binding to LPS intracellularly. So far, only in vitro pull-down data support this (but this would not be surprising since Gal3 is also present extracellularly). The interaction of Gal3 with the Rags under LPS-induced conditions was shown convincingly. However, whether this occurs via intracellular binding of LPS to Gal3 and thus whether Gal3 is “sensing” LPS remains obscure. The authors should either perform experiments to further address this issue (eg use of Gal3 mutants), or modify their conclusion as to the “sensor” nature of Gal3. The data are supportive of LPS-induced mTORC1 activation via Gal3 binding to Rags/Ragulator but do not provide convincing data on intracellular LPS binding to Gal3, hence the LPS sensor status of Gal3 is tenuous.

Authors: In addition to Supplementary Fig. 2a, Fig. 2e presents the evidence that the Gal3 level affects the activation of mTORC1.

As regard to Gal3 binding to LPS intracellularly. We showed lysosomal colocalization of LPS and Gal3 in Fig. 4a. Plus, Fig. 4d shows that LPS affects Gal3 interaction with Rag GTPases and Ragulator, Fig. 3d shows that LPS activates the mTOR signaling pathway only in the presence of Rag GTPases and Gal3. In addition, studies by others (PMID: 34301890) also showed that cytosolic LPS binds to intracellular Gal3. In light of the reviewer’s comments, we have highlighted these evidence and arguments in the revision.

Fig. 4f:

Line 275-276: “we next investigated whether the interaction of galectin-3 with RagA/RagC and Ragulator influences the nucleotide state of Rag GTPases.”

Line 289-290: “...suggesting that the interaction of galectin-3 with RagA/RagC is functionally linked to the nucleotide state of RagA and the generation of active Rag

GTPases.”

The experiments in Fig 4f examines how Gal3 binds to the Rags, ie their conclusion is that Gal3 prefers to bind to active Rags by virtue of their nucleotide state. Thus, Lines 275-276 should be modified. They did not address whether the Gal3 binding influences the nucleotide state of Rags. This is the reason why this reviewer inquired whether Gal3 absence could decrease Rag-GTP loading. The authors did not address this inquiry, however.

Authors: As we repeatedly stated in our manuscript, our overall argument is that Gal3 binds to Rag GTPases and Ragulator to facilitate the interaction of these components, leading to the activation of Rag GTPases, i.e., the transition of the nucleotide state. To comply with the reviewer, we have modified the texts as “We next investigated the relationship between the interaction of galectin-3 with RagA/RagC and Ragulator and the nucleotide state of Rag GTPases”.